# Are researchers following best storage practices for measuring soil biochemical properties?

Jennifer M Rhymes[1,2] *, Irene Cordero[1] *, Mathilde Chomel[1], Jocelyn M Lavallee[1,3], Angela L Straathof[1,4], Deborah Ashworth[1], Holly Langridge[1], Marina Semchenko[1,5], Franciska T de Vries[1,6], David Johnson[1], Richard D Bardgett[1]

[1] Department of Earth and Environmental Sciences, Michael Smith Building, The University of Manchester, Oxford Road, Manchester, M13 9PT, UK

[2] Environment Centre Wales, Bangor University, Deiniol Road, Bangor, Gwynedd, LL57 2UW, UK

[3] Department of Soil and Crop Sciences, Colorado State University, Fort Collins, CO, 80523, USA

[4] Ontario Soil and Crop Improvement Association, 1 Stone Road West, Guelph, ON N1G 4Y2, Canada

[5] Institute of Ecology and Earth Sciences, University of Tartu, Lai 40, Tartu, 51005, Estonia

[6] Institute for Biodiversity and Ecosystem Dynamics, University of Amsterdam, PO Box 94240, 1090 GE, Amsterdam, the Netherlands

* J M Rhymes and I Cordero contributed equally to this work.

**Abstract**

It is widely accepted that the measurement of organic and inorganic forms of carbon (C) and nitrogen (N) in soils should be performed on fresh extracts taken from fresh soil samples. However, this is often not possible, and it is common practice to store samples (soils and/or extracts), despite a lack of guidance on best practice. We utilised a case study on a temperate grassland soil taken from different depths to demonstrate how differences in soil and/or soil extract storage temperature (4 °C or -20 °C) and duration can influence sample integrity for the quantification of soil dissolved organic C and N (DOC and DON), extractable inorganic nitrogen ($NH_4^+$ and $NO_3^-$ ), and microbial biomass C and N (MBC and MBN). The appropriateness of different storage treatments varied between topsoils and subsoils, highlighting the need to consider appropriate storage methods based on soil depth and soil properties. In general, we found that storing soils and extracts by

freezing at -20 °C was least effective at maintaining measured values of fresh material, whilst refrigerating (4 °C) soils for less than a week for DOC/DON, up to a year for MBC/MBN, and refrigerating soil extracts for less than a week for $NH_4^+$ /$NO_3^-$ did not jeopardise sample integrity. We discuss and provide the appropriate tools to ensure researchers consider best storage practice methods when designing and organising ecological research involving assessments of soil properties related to C and N cycling. We encourage researchers to use standardised methods where possible and to report their storage treatment (i.e. temperature, duration) when publishing findings on aspects of soil and ecosystem functioning. In the absence of published storage recommendations for a given soil type, we encourage researchers to conduct a pilot study and publish their findings.

Keywords: Soil, Sample Storage, Microbial Biomass C and N, Analytical Biogeochemistry

## 1    Introduction

Biogeochemical cycles involve the turnover of essential nutrients between different organic and inorganic forms. For carbon (C) and nitrogen (N), many of these steps occur in the soil environment and hence the evaluation of different chemical forms of nutrients in soil is crucial to understand the recycling of nutrients and ecosystem functioning (Barrios, 2007; Datta, 2020; Robinson et al., 2014). It is therefore integral that researchers consider each factor that can impact accurate and reliable analytical measurements, which can include sampling procedures (e.g. strip removal of turf), transport (e.g. transport length and temperature), storage (e.g. temperature), preparation for analysis (e.g. sieving mesh size and when samples are sieved) and analytical methods (e.g. temperature, shaking times and filter types). Here we focus solely on sample storage. While most soil biogeochemical analyses should ideally be carried out on fresh samples immediately after sampling (ISO18400-102:2017, 2017), this is not always possible due to the number of samples taken and the analytical procedures exceeding human and/or instrumental capabilities. In these cases, it is common practice to store samples for future analysis. These can include freeze drying, air drying, freezing and refrigerating samples, and the method is typically chosen dependent on the analysis in question and time in which analysis can take place.

Soil extraction procedures are commonly used to quantify different biochemical parameters in soils. Typically, such procedures shake soils with a high soil weight-to solution volume ratio and separate the solution phase from the solid phase by centrifugation and/or filtration (Kachurina et al., 2000). This process poses further storage opportunities for future analysis, irrespective of how soils were initially stored. However, recommendations for both soil and/or soil extract storage vary substantially, and little is known about the impact storage methods may have on sample integrity.

Dissolved organic C and N are commonly extracted from soils with water (Forster, 1995). However, in cases where inorganic N is also being quantified, concentrated salt extractions, such as KCl, are used to evaluate 'plant available' N (Forster, 1995; Jones and Willett, 2006; Keeney and Nelson., 1982). Methodological factors for both extraction types differ substantially (Jones and Willett, 2006; Ros et al., 2009).

Many comparative studies exploring the impacts of methodological factors overlook soil and/or extract storage temperatures and duration, and when these were considered, few storage possibilities were taken into account (Table 1). For example, a meta-analysis exploring methodological factors that impact soil extractable organic N did not account for soil or extract storage duration, despite showing impacts from soil storage temperatures and soil extract temperatures (Ros et al., 2009). Nevertheless, while recommendations for storage of soil, as well as water and KCl extracts are reported, they are in many cases vague with
no indication as to when samples deteriorate beyond usability, highlighting the need for more comparative studies.

Table 1 –Summary of different recommendations for storage of soil or extract samples to measure soil nutrients found in the literature. This summary is non-exhaustive. The term "not applicable" under soil type refers to studies that were not based on comparative studies and therefore were not carried out on a soil type. The term "not provided" refers to comparative studies that do not describe the soils explored in the methods.

| Variable evaluated | Extractant used | Soil type | Study | Recommendation based on | Storage methods explored | Storage recommendations | Limitations |
|---|---|---|---|---|---|---|---|
| Water extractable organic carbon | H2O | Not applicable | Gregorich and Carter, 2007 | No evidence provided | | Minimal time, refrigerated maybe ideal | |
| | | Three soils: Loam, sandy loam, sandy clay | Rees and Parker, 2005 | Comparative study | Extractant 4 °C, -18 °C and room temperature | Store extracts at 4 °C for 1 week. / Store extracts frozen at -18 °C for 3 months / Do not store extracts at room temperature | |
| | | Yolo loam, Typic, Xerorthents family (USDA classification). | Rolston and Liss, 1989 | Comparative study | Soils stored as air dried and frozen at -10 °C | Store soils at -10 °C for two months if storage is required | Only one storage length was explored |

| | | | | | | | |
|---|---|---|---|---|---|---|---|
| Plant available N | KCl | Not applicable | Heffernan, 1985 | No evidence provided | | Store extracts at -18 °C indefinitely | |
| | | Not provided | Li et al., 2012 | Comparative study | Air dried soils compared to fresh. Extracts at 4 °C, -18 °C and room temperature (25 °C) | Do not air dry soils and extract as soon as possible. Store extracts at -18 °C and analyse as soon as possible | Storage length explored up to 6 weeks only |
| | | Unclear. Cambisol, podzol and/or gleysol. | Jones and Willett, 2006 | Comparative study | Air dried soils compared to fresh. Extracts at 4 °C and -20 °C | Carry out soil extractions within 24 hours of collection. Store extracts for days in the refrigerator. Store extracts at -20 °C for moths | Results from the extract storage test are not clearly shown. Vague recommendations made for extract storage length which could be open to different interpretation |
| | | Not applicable | Gregorich and Carter, 2007 | No evidence provided | | Minimal time, refrigerated maybe ideal | |
| Microbial biomass | K2SO4 | Arable sandy loam soil, grassland orchard soil and mixed forest soil with high organic carbon content | Černohlávková et al., 2009 | Comparative study | Soils stored at 4 °C, -20 °C and air dried | Store sieved soil at 4 °C for up to 8 weeks | |
| | | Not applicable | Vance, Brookes and Jenkinson, 1987; Beck et al., 1997; Coleman, Callaham and Crossley Jr, 2017 | No evidence provided | | Store extracts indefinitely at -18 ºC | |
| | | Agricultural mineral | Stenberg et al., 1998 | Comparative study | Soils at 2 °C and -18 °C | Store soils at -18 °C for up to 13 months | Extracts were also frozen at -20 °C until analysed with no account for storage length |
| | | Not applicable | Gregorich and Carter, 2007 | No evidence provided | | Minimal time, refrigerated but not frozen | |

Microbial biomass C and N are commonly quantified using fumigation-extraction methods (Brookes et al., 1985; Vance et al., 1987). In their classic paper, Vance et al. (1987) recommended that $K_2SO_4$ extracts should be analysed immediately, and where this is not possible, stored for up to 2 weeks at 1-2 °C. However, these authors did not give any recommendations for storing soil samples prior to extraction, which is also commonly practiced. Nonetheless, many studies have since modified the Vance et al. (1987) and Brookes et al. (1985) methods, which has led to substantial variation in practice and storage of soil and extracts (Table 1). To the best of our knowledge, only the recommendations of Stenberg et al. (1998) and Černohlávková et al. (2009) were based on comparative studies of different storage methods, whereby sample integrity was best preserved when fresh soils were stored at -20 °C for up to 13 months or when soils were stored at 4 °C for up to 8 weeks, respectively. Despite these findings, Stenberg et al. (1998) still stored the extracts of both soil storage treatments at -20 °C until analysis and made no account for storage length.

We highlight that recommendations for storage methods are vague and that there is a lack of comparative studies to determine best storage practices for the quantification of soil DOC, DON, inorganic nitrogen and microbial biomass, which are all commonly measured in ecological studies considering aspects of soil and ecosystem functioning. We also explored common practices across different laboratories with an online survey (details provided in Supplementary 3.4), which suggests that storage of both soils and extracts is common practice (Fig. S1). Generally, the storage of soil was done at 4 °C for a short period of time (<1 week), while extracts were stored at -20 °C and for longer (> 4 weeks, Fig. S1). Nonetheless, storage methods varied significantly, highlighting the need for common protocols to standardize methods across laboratories. In our case study, we chose to explore refrigerating and freezing storage practices instead of other storage methods (e.g. air drying or freeze drying) because there is significant evidence to suggest that other methods are unsuitable for the variables we measure. For example, air drying soils has a strong effect on C and N pools, probably due to microbial death and nutrient release upon drying and rewetting (Jones and Willett, 2006; Kaiser et al., 2001; Li et al., 2012; Rolston and Liss, 1989). Additionally, freeze-drying is also known to have a strong effect on nutrient pools, as the chemical, physical, and physiological stresses inflicted by freeze-drying can kill soil microbes, releasing the microbial compounds into the soil (Islam et al., 1997).

In this commentary, we report a study that aimed to identify the best practice methods for storage of soil or soil extracts for the analysis of soluble pools of C and N and microbial biomass in soil. The study, which was based on both topsoil and subsoil of a well-characterised experimental grassland site (Leff et al., 2018; De Long et al., 2019), serves to demonstrate how different, widely used storage methods can affect sample integrity. It also provides the tools required by researchers to determine best storage practice for their own studies, given that optimal storage methods will vary across different soils and ecosystem types. We encourage researchers to carry out their own pilot studies, for which our study provides an example and guidelines for.

## 2    Case Study

## 2.1 Brief description of methods and experimental design

Our study aimed to determine best practice methods for storage of soil or soil extracts for the analysis of dissolved organic C (DOC), dissolved organic N (DON), inorganic N ($NO_3^-$, $NH_4^+$), and soil microbial biomass (MBC and MBN). This was tested on both topsoil (0-20 cm) and subsoil (20-30 cm) of a brown earth (Cambisol) taken from a well-studied experimental grassland site (De Long et al., 2019; Leff et al., 2018; Table S1), which is representative of typical permanent grasslands used for livestock production across the UK and parts of Europe (Rodwell, 1992). We designed a full factorial experiment with both topsoil and subsoil, two different types of stored samples (soil or extract), and two different storage temperatures (4°C or -20 °C), replicated five times. We evaluated four different types of extracts: water, 1 M KCl, fumigated 0.5 M $K_2SO_4$ and unfumigated 0.5 M $K_2SO_4$; at 12 different time points: 1, 3, 7, 14, 21, 28, 57, 85, 113, 169, 281 and 430 days after sampling. Additionally, we measured and analysed the four different extracts immediately after soil collection (fresh sample), to use as a 'baseline' comparison value (amounted to 1,952 extractions in total).

All statistical analyses were carried out in R Version 3.6.1 (R Core Team, 2019). In order to standardize the relative change of each variable measured for each soil type, storage type and storage length to the measurements made immediately on the fresh samples, we calculated a ratio for each corresponding replicate with the below equation:

$$Relative\ change\ = \frac{Measured\ variable\ for\ each\ treatment}{Measured\ variable\ from\ fresh\ sample}$$

Mixed-effects models were performed for each measured variable with the lme4 package (Bates et al., 2018) to test the effects of fixed factors (soil type, storage type and storage length) and random factor (replicate) and their interactions on the calculated relative change ratio from fresh samples (baseline). Predicted fitted values from the multi-level model were calculated with *predictInterval* with the *merTools* package (Frederick, 2019).

Similarity between fresh samples (baseline) and soil storage treatments was determined when the upper or lower limit of the predicted fitted value confidence intervals fit within 20% positive and negative variance from fresh samples (baseline); we refer to these as similarity limits (Rita and Ekholm, 2007; Wallenius et al., 2010). For further detail on sample collection and preparation, storage treatments, extraction procedures and statistical analysis please read our full study description in the supplementary material provided.

## 2.2 Results

Overall, we found significant impacts of storage method and duration of both topsoil and subsoil on several response variables. In topsoil, we found that refrigerating soils, freezing extracts up to 430 days, and refrigerating extracts up to 10 days

successfully maintained similar DOC concentrations to those from fresh samples (Fig. S2a). Freezing soils always resulted in
dissimilar DOC concentrations to fresh samples regardless of storage duration. DOC concentrations increased immediately
after freezing and continued to increase over time. With regard to subsoil, freezing soils, refrigerating extracts up to 430 days,
and refrigerating soils up to 8 days successfully maintained similar values to fresh samples, but freezing extracts led to
significantly different DOC concentrations compared to fresh samples (Fig. S2a).

DON concentrations in water extracts from topsoil stored for up to 281 days in the refrigerator or freezer were similar to those
of fresh samples (Fig. S2b). DON concentrations in stored topsoils were unaffected by refrigerating soils for up to 60 days,
while freezing topsoils changed DON concentrations relative to fresh samples throughout the experiment. DON concentrations
increased immediately after freezing and continued to increase with storage duration, as observed for DOC. For subsoils,
refrigerating soil samples up to 3 days was deemed to be the only storage method to yield similar results to the fresh samples,
with all other storage treatments of any duration yielding dissimilar results (Fig. S2b). DOC extracts from blank (ultrapure
water) samples used for blank corrections only differed with storage length when stored in the refrigerator, where DOC
concentrations increased with increased storage length doubling its concentration after 430 days (Fig. S3).

All storage types were inappropriate for analysis of extractable $NO_3^-$ in both soils, apart from refrigerating extracts up to 5
days and 42 days for topsoil and subsoil, respectively (Fig. S4a). There were no storage methods that were deemed appropriate
for measuring extractable $NH_4^+$ in subsoils (Fig. S4b). However, refrigerating soils and extracts, and freezing extracts up to
135, 141 and 430 days from topsoil yielded $NO_3^-$ concentrations similar to those in fresh samples. By contrast, freezing soils
was not appropriate for any storage length in topsoil (Fig. S4b).

Subsoil MBC did not differ from fresh soil when soils were frozen for up to 430 days (Fig. S5a), while every other storage
treatment did within just one day of storage. By contrast, MBC in topsoil was similar to fresh samples in refrigerated soils,
refrigerated extracts and frozen extracts up to 430 days, and in frozen soils up to 75 days. However, separate evaluation of the
fumigated and unfumigated samples revealed differences (Fig. S5b, c). Fumigated extracts were comparable to fresh samples
in all storage methods for topsoil, but only when soil was stored (either in the refrigerator or frozen) for subsoils (Fig. S5b).
For both soils, TC generally decreased in the fumigated refrigerated extracts with long storage times (starting after 3 months
of storage), at least for most replicates. Unfumigated extracts were only comparable to the fresh samples in topsoil if the soil
was refrigerated, while all storage methods were comparable to the fresh samples up to 430 days in subsoil (Fig. S5c).

MBN data were comparable to the fresh measurements for both soils and all storage types, except for the frozen soil from
subsoils (Fig. S6a). As for MBC, fumigated and unfumigated extracts did not follow the same trend. TN in fumigated extracts
was comparable to the fresh for both soils and for all storage times (Fig. S6b). However, TN in unfumigated extracts showed
more variability (Fig. S6c). Storing extracts was an appropriate storage method for both soils, but storing subsoil only deemed
appropriate when stored in the refrigerator. Freezing soil led to an immediate increase of TN in both soil depths.

## 2.3 Discussion

### 1.1 Storing soils

Refrigerating sieved soils for up to 3 days was deemed the most appropriate storage method for the quantification of DOC and DON in both topsoil and subsoil. Rolston and Liss (1989) recommended to freeze soils if storage is required; by contrast, for the quantification of DOC, we found freezing sieved soils to result in the largest shifts in DOC and DON concentrations. Topsoil DOC and DON concentrations increased beyond comparison with fresh samples within just one day of freezing. Increases in DOC after storing soils in the freezer have previously been reported (Kaiser et al. 2001, Ross & Bartlett 1990), as observed here in our study. A combination of factors associated with increasing labile C and N availability from a freeze-thaw cycle were likely to have contributed to these results, including the release of DOC and DON from microbial death (Černohlávková et al., 2009), a change in soil structure (van Bochove et al., 2000) and root decomposition (Tierney et al., 2001). However, shifts in DOC and DON concentrations also persisted with longer storage length implying that there are other factors contributing to these shifts beyond those related to the freeze-thaw process.

Storing refrigerated soils was the least appropriate method for the quantification of extractable N, as $NO_3^-$ concentrations increased considerably and continued to increase with storage time in both topsoil and subsoil. This was likely due to a combination of: 1) the inability of refrigerated temperature to stop mineralisation (Tyler et al., 1959); 2) increased rates of N mineralisation after sieving (Hassink, 1992); and 3) reduced $NO_3^-$ uptake by plants due to plant removal. This is supported by our observed decrease in soil DON concentration.

In general, refrigerating soils was an appropriate storage method to evaluate MBC and MBN, in line with the findings of Černohlávková et al. (2009). However, microbial biomass may be calculated inappropriately as an artefact of divergent changes in fumigated and unfumigated samples incurred from storage treatments and therefore requires both fumigated and unfumigated extraction samples to meet similarity limits. We found that freezing soils to measure MBC was acceptable (up to 75 days for topsoil and 430 for subsoil), but not for MBN (although acceptable for topsoil up to 430 days). We therefore deemed freezing soils as an inappropriate storage method for quantifying microbial biomass because the subsoils were jeopardised by freezing soil samples to quantify MBN. Our recommendations are therefore contrary to those made by Stenberg et al. (1998), despite finding similar results for MBC. We found that freezing soils generally increased extractable C and N concentrations in unfumigated extracts but did not affect concentrations in fumigated samples. This suggests freezing caused some microbial death (Černohlávková et al., 2009) precluding reliable quantification of microbial biomass using fumigation. Refrigerating soil for the quantification of C in unfumigated soil was appropriate for up to 430 days, yet deemed inappropriate for N in topsoil. Generally, topsoils are susceptible to more storage-related changes than mineral soils (Lee et al., 2007), as a result of their greater microbial biomass. In this instance, topsoil had 720 % greater MBC and 390% greater MBN than subsoil

making them more susceptible to nutrient turnover (Schnecker et al., 2015), where increased mineralisation from sieving may have contributed to this (Hassink, 1992).

## 1.2 **Storing Extracts**

Although refrigerating extracts for the quantification of DOC was appropriate for up to 10 days, we identified an underlying issue with longer periods of this storage method as blank extracts accumulated DOC over time when stored in the refrigerator.

We were unable to determine what may have caused this, but it highlights the importance in considering the implications of every methodological step within a procedure and the necessity to include blanks for analysis. For example, the potential leaching of DOC from the polypropylene tubes where the extracts were stored could have contributed to this as it has been demonstrated that plastic can leach DOC into the water, even if kept in the dark and under sterile conditions (Romera-Castillo et al., 2018). Freezing the sample might have prevented this leaching. In support of and in line with recommendations made

by Rees and Parker (2005), we found that freezing topsoil water extracts was an appropriate storage method throughout the duration of the experiment; however, this was not the case for subsoil.

Filtering extracts before storage can also pose issues with sample preservation. When extracts are filtered through pore sizes larger than 0.22 μm, the sample is not sterilised, resulting in biologically active extracts that are susceptible to microbial transformations of C and N (Ghuneim et al., 2018; Wang et al., 2007). This issue is likely to have also contributed to $NO_3^-$

losses in refrigerated 1 M KCl extracts, as denitrification is accelerated in anaerobic conditions, and the decreasing C trend with longer storage in both refrigerated and frozen fumigated and unfumigated 0.5 M $K_2SO_4$ extracts. This is supported by observations of fungal growth in many $K_2SO_4$ extracts after three months of extract refrigeration. Consequently, refrigerating extracts for up to 5 days proved to be the only viable option for the quantification of extractable $NO_3^-$ for both topsoil and subsoil, contradictory to reports that recommend freezing KCl extracts for months (Jones and Willett, 2006; Li et al., 2012),

or in some instances indefinitely (Heffernan, 1985). Furthermore, storing fumigated extracts of subsoils either in the refrigerator or freezer were also not appropriate storage methods for the quantification of MBC, despite recommendations to refrigerate extracts for up to 1-2 weeks (Vance et al., 1987) or at -18 °C for an indefinite period (Beck et al., 1997). However, freezing samples did not significantly affect the concentration of N in fumigated or unfumigated samples, and thus frozen extracts was a suitable storage method to measure MBN. Due to the potential for freeze-thaw cycles to impact sample

biogeochemistry (Černohlávková et al., 2009), it is important to also consider and be consistent with the freeze/thaw procedure, such as the position in which extracts are frozen (vertical or horizontal placement of tubes) or under which conditions extracts or soils samples are thawed (e.g. thawing soils over night at 4°C or extracting frozen soil immediately with the solution). Although we found it to be generally unadvisable to store soil extracts, this procedure may be appropriate if samples are sterilised or stored in conditions that completely halt microbial activity, which is likely to be one of the main mechanisms

leading to changes in nutrient concentrations. For example, adding acid prior to storage (Zagal, 1993) or microbial inhibitors

(Rousk and Jones, 2010) has been suggested, but this may not be compatible with instrumentation and the quantification of inorganic nutrient pools, and requires further investigation.

## 1.3    Key findings

Our study provides strong evidence that storing soils and extracts can have significant consequences for the quantification of soluble C and N pools of relevance to key ecosystem processes. These findings are important given increasing emphasis on the need to understand soil processes as regulators of ecosystem services (Coe and Downing, 2018; Dangi, 2014), and calls for standardised and robust indicators of soil health made in recent policy interventions (DEFRA and EA, 2018), where consistency in protocols across studies and measurements is essential. Overall, we found significant impacts of storage method and duration demonstrating that it is generally not advisable to store soils or soil extracts. Nonetheless, through appropriate experimental design we were able to determine a limited range of storage type and storage duration recommendations for both topsoil and subsoil (Table 2). We found that storing soil and extracts by freezing at -20 ℃ was generally least effective at maintaining measured values of fresh material. Appropriate storage recommendations include refrigerating (4 °C) brown earth soils for less than a week for DOC/DON and up to a year for MBC/MBN, and refrigerating extracts for less than a week for $NH_4^+/NO_3^-$.Table 2. Storage method recommendations for both temperate topsoil and subsoil. Dark grey denotes inappropriate storage methods for a specific analysis. Light grey denotes appropriate storage method, where storage length is annotated. Where storage length is annotated as 430 days we are unable to advise storage length beyond this due to the length of the experiment. Storage methods are deemed appropriate: 1) if the storage method does not compromise the sample integrity (defined as stored samples yielding soil parameter values within 20% similarity limits to fresh samples) for both topsoil and subsoils explored; and 2) where the same extractant type is used to measure different parameters, the storage method does not compromise the integrity of each parameter measured.

| Measured Variable | Topsoils | | | Subsoils | | |
|---|---|---|---|---|---|---|
| | Extractable Dissolved Organic N and C | Inorganic N | Microbial Biomass | Extractable Dissolved Organic N and C | Inorganic N | Microbial Biomass |
| **Extractant** | Water | KCl (1 M) | $K_2SO_4$ (0.5 M) | Water | KCl (1 M) | $K_2SO_4$ (0.5 M) |

| Storage Type | | | | | | | | |
|---|---|---|---|---|---|---|---|---|
| Soil | 🧊(fridge) | <1 month | | 430 days | <1 week | | 430 days |
| | ❄(freeze) | | | | | | |
| Extract | 🧊(fridge) | 10 days | <1 week | | | | |
| | ❄(freeze) | 430 days | | | | | |

It is commonly assumed that any changes to soil biochemistry from storage methods will occur equally for all samples. Here, we provide evidence to show that changes do not occur equally which could have major implications for the findings of ecological studies. We did not investigate the mechanism behind the different responses to the storage treatments, but it could be due to differences in physical and chemical properties of soils at different depths, and lower substrate availability with increasing depth (Bardgett et al., 1997) resulting in smaller microbial biomass (Lavahun et al., 1996), reduced microbial activity (Schnecker et al., 2015), and a decreased capacity for substrate utilisation (Kennedy et al., 2005). As a result, any treatment that affects soil properties has the potential to also affect the response of soils to storage. Even if sample biochemistry changes immediately as a result of storage but subsequently remains stable over storage time, in our study this effect varied between the two soil depths. Therefore, even if the research question is to compare between treatments applied to the same soil type, strict storage limits should still be explored and followed. We suggest that all samples are stored under the same conditions that allow the preservation of samples from the soil type, site and/or treatment with the highest sensitivity to storage. This can be determined through rapid review methods and/or pilot studies which we discuss in section 3. We would also like to note that due to the high temporal variability that the temperate soils explored experience, there is the potential that storage methods could also impact sample integrity differently depending on when the samples were collected. Understanding the mechanisms responsible for jeopardising sample integrity under different storage methods will help determine the best storage methods for the time in which samples are collected (e.g. season), soil type and depth.

## 3    How to determine best storage practice for your experiment

The case study findings highlight how integral it is to consider best storage practice for soil analysis in any study/experiment, this includes studies exploring one or more soil types, site locations and/or treatment manipulations. We provide a step by step systematic flow chart to determine best storage methods for soil and soil extracts (Figure 1).

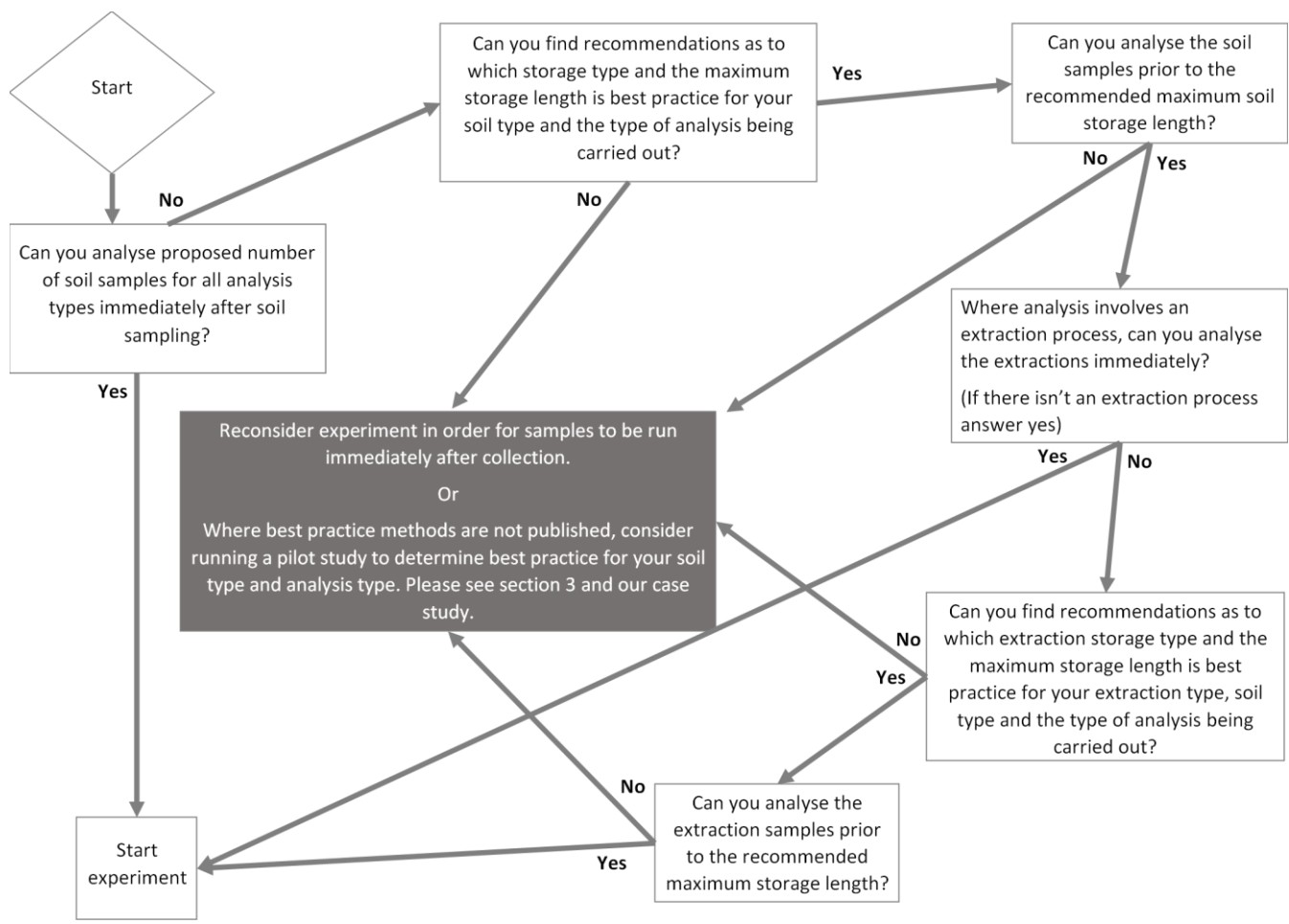

Figure 1 Schematic flowchart depicting necessary steps to determine best storage practices for soil and soil extracts in ecological studies.

Where there are publications outlining best soil storage practices, ensure recommendations are based on comparative studies carried out on the correct soil type. Where published recommendations are not found, we advise researchers to carry out a targeted pilot study using less extensive yet similar approaches to that outlined in our case study. We identified key considerations that need to be made in Table 3 to ensure that comparisons between storage methods tested for are appropriate for determining best storage practices.

Table 3. Considerations to be made and their associated issues and recommendations when designing a pilot study.

| Consideration | Issues | Recommendations |
|---|---|---|
| Soil type | Responses to storage methods vary between soil types. | When working with different soil types we recommend making comparisons between storage methods for each soil type. |
| Time points | Limited by resources. | Choose a reasonable set of time points within resource limitations. Include best- and worst-case scenario for the timeframe that you typically need to analyse samples after collection. |
| Scaling | Pseusoreplication, reproducibility. | Do not scale your soils or extracts for storage up (bulk storage) or down. The same weight or volume of soil or extract must be stored separately for each storage treatment and time point as the one planned for the main experiment. |
| Extraction matrix | Each extraction matrix will respond differently to each storage method. | Storage methods for each extraction matrices should be considered separately. |
| Extraction methods | Extractant volumes, shaking times, centrifugation times and filter types can influence measurements. | Use the same extraction methods throughout all storage treatments and for baseline measurements. Where possible utilise standardised methods (e.g. Halbritter et al., 2020). |
| Baseline[1] | Without reliable baseline measurements conclusions on best storage practice cannot be made. | Double the number of replicates for this time point (day 0) and ensure analysis is carried out immediately after soil collection. |
| Replicates | Heterogeneity. | Generally, we recommend as many replicates as one can afford to have but recommend no fewer than 4 as suggested by Jones and Willet (2006). For more guidance on choosing the number of replicates, we advise researchers to utilise the sample size calculator formula from Cochran and Cox (1957), p. 20. |

| | Pseudo replication. | Do not take replicates from same sampling location, ensure replicates capture the range of soil variability. |
|---|---|---|
| | | Do not store soils or extracts in bulk. |
| Blanks | Some storage vessels can leach DOC. | Ensure you have a minimum of three replicate blanks for each extract type, storage method and time point. |
| Setting your upper and lower similarity limit[2] | Heterogeneity. | Replicate baseline measurements of the same soil sample will indicate the level of variation in measurements due to subsampling, handling (e.g. filtering) and instrument (e.g. calibration and accuracy) effects. This variation can inform the decision on the similarity limits or you can choose to accept a 10% or 20% upper and lower limit. |
| Deciding on the best storage practice | When working with more than one soil type. | Samples should be subjected to the same storage method and length that is deemed appropriate for all soil types. For example, we found that it is appropriate to store brown earth subsoils at 4 °C for less than a month to quantify DOC/DON by water extractions (Table 2). However, for brown earth topsoils we found that this storage method was only appropriate for soils stored for less than a week, thus limiting the storage length to one week for both soil types. |

[1]Soil measurements immediately after soil collection, not subjected to any storage method

[2]The negative and positive percentage variance from baseline measurements accepted between baseline and storage method measurements to deem storage method appropriate

Where possible, we strongly advise researchers to publish pilot studies (as a minimum within supplementary materials) to ensure approved methods are adopted by the wider ecological community and for the future synthesis in development of a standardised practice handbook for all soil types.

**4    Improving method reporting**

Comprehensive reporting of storage practices based on pilot studies and published recommendations in the literature are important for improving storage practices amongst the ecological community. It also poses new opportunities for meta-analyses and syntheses to explore and determine effective and accurate methodological practices quantifying ecological processes. Furthermore, this allows for context dependencies in the effects and responses to each practice to be investigated (e.g. soil type). It is therefore integral for researchers to report sampling locations with coordinates, detailed information on soils (including World Reference Base for Soil Resources WRB soil type and characteristics), detail modifications made to any referenced methods and to report the storage methods used. With focus on storage methods, we recommend that both the storage duration and basis for using a particular storage method is detailed. For example, "Extractions were carried out on soil samples immediately after soil collection. Soil extract samples were stored at 4°C for one week as recommended by our own pilot study reported in supplementary material."

## 5    Conclusions

Our results demonstrate that it is generally not advisable to store soils or soil extracts when assessing soluble C and N pools and microbial biomass. We also show that the appropriateness of different storage treatments varied between topsoil and subsoil, suggesting that appropriate storage methods need to be tailored for different soils. However, we recognise that it is not always possible to avoid storing soils and therefore recommend using the tools provided to determine best practice.

We stress that researchers must also consider other practices beyond just storage (e.g. sieving samples, transport, extraction procedures) as each methodological step between sample collection and analysis can introduce errors to measurements that are intended to be field representative. We encourage researchers to utilise standardised methods where possible (see e.g. Halbritter et al. 2020) and to follow best storage practices for specific soil types to allow reliable comparison of data from different studies. Given the potential for storage treatment to affect results, we also urge researchers to report detailed information about their storage treatment (i.e. temperature, duration) and the basis for the chosen treatment when publishing findings. In the absence of published storage recommendations for a given soil type, we encourage researchers to conduct a pilot study and publish their findings. This will allow for future synthesis and development of a comprehensive handbook for standardised methods for soil and/or soil extract storage as many published standardised methods currently give unsubstantiated advice.

## 6    Data Availability

Data is available upon request of the authors.

## 7    Author Contributions

All authors contributed equally towards the conceptualisation of the study. I Cordero, H Langridge, D Ashworth, J Lavallee, A Straathof, M Chomel and J Rhymes carried out fieldwork and laboratory analysis. M Semchenko and M Chomel provided statistical analysis support. I Cordero and J Rhymes co-ordinated the project, analysed the data, prepared the manuscript, responded to reviewers' comments and edited manuscripts with contributions from all co-authors. RD Bardgett, D Johnson, F De Vries, M Semchenko and A Straathof acquired funding for the project.

## 8 Competing Interests

The authors declare that they have no conflict of interest.

## 9 Acknowledgments

We gratefully acknowledge JR de Long and E Fry for contributions towards the experimental design. We also thank all of the Soil and Ecosystem Ecology Laboratory members at The University of Manchester for their help and support in the lab. This project was supported by a PDRA Research Fund from the Department of Earth and Environmental Sciences, The University of Manchester, and awards to RDB (NERC Soil Security: NE/M017028/1, and BBSRC: BB/I009000/2), FTdV (BBSRC David Phillips Fellowship: BB/L02456X/1), DJ (NERC Soil Security: NE/M017028/1, and the N8 AgriFood programme) and IC (Ramon Areces Foundation Research Fellowship, and BBSRC Discovery Fellowship: BB/S010661/1).

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
