# Peer review of "Are researchers following best storage practices for measuring soil biochemical properties?"

_SOIL, 2020_

## Referee Comment (RC1) · Anonymous Referee #1 · 22 Dec 2020

In this short communication the authors address the lack of evidence-based recommendations for storage of soil samples and extracts prior to analysis of organic and inorganic carbon (C) and nitrogen (N) and microbial biomass C and N. Based on literature and an online survey, the authors demonstrate that samples and extracts are stored in a multitude of ways. They further provide a case study in which they demonstrate the effects of sample and extract storage on measurements of organic, inorganic and microbial C and N, further stressing the need for standardisation of protocols. The manuscript offers recommendations for sample and extract storage for the investigated methods, a flow-chart to guide researchers deciding on the most appropriate storage approach for their experiments and recommendations for reporting of the methods

adopted for scientific publications. The results of the case study are a welcome reference for future research. In particular, the case study demonstrate that storage methods do not affect samples of different soil types (top vs subsoil) equally. This finding goes against the common assumption that storage affect all sample types similarly.

This short communication is relevant for the international science community, and fits well in the scope of SOIL. The introduction outlines the motivation of the objectives clearly, and the necessary details of the case study methodology and results are given in the supplementary material. The text is well written and pleasant to read. I have a few suggestions for corrections. Tables and figure are mostly supportive (particularly table 2). I have some suggestions to further clarify the text in tables 1 and 3. Also, I wonder whether the message the authors want to convey in section 4 with Table 4 isn't better served by integrating its information into the text. Lastly, I would ask the authors to consider including Fig S6 into the main text. I outline my general and minor comments in more detail below.

General comments The authors conducted an online survey and reflect on the outcome in L65 and further. Adding few lines about the nature of the survey and how representative the responses are would increase the value of its outcome in the authors argumentation. Also, inclusion of Figure S6 into the introduction would aid the reader follow the narrative, or at least place it first in the Supplementary materials. For Figure S6, I recommend the authors to clarify the meaning of the colours, explain the abbreviation of RT, and consider scaling the size of the arrows to the % of cases.

I miss the mentioning of other storage methods than those considered by the authors in their case study. L73 states the authors considered widely used storage methods, but it does not become clear to me how the selected methods compare to alternatives, for example drying. I recognize that a full evaluation is beyond the scope of this short communication, yet adding a few examples could bring the proposed manuscript into a broader perspective. For example, storage and the achievability of soil samples are important considerations when choosing indicators for monitoring efforts (Ritz K, Black HIJ, Campbell CD, et al (2009) Ecol Indic 9:1212–1221. https://doi.org/10.1016/j.ecolind.2009.02.009).

While sections 2 and 3 give helpful directions to soil scientists, I find section 4 less strong. Table 4 presents best reporting practices, yet the section does not mention why the listed requirements are important, nor gives a motivation for providing these recommendations. Also, I would recommend the authors to remove table 4, and instead list the four reporting recommendations in the text in section 4. To convey the best reporting practice, I don't think it is necessary to provide examples of poor reporting, and giving examples of good reporting are easier to read when listed in the text.

Minor issues: L43: "that have considered these have taken into account few..." makes it easier to read

Table 1: Column 6 "storage methods explores", remove the word 'only' in the listed examples. Yolo loam: what soil type reference was used, e.g. WRB soil types? $H_2O$ – Rolson and Liss: at what temperatures were the soils stored frozen? Plant available N – Soil Type: what is the difference between not applicable and not provided? Plant available N – Jones & Willett 2006: can the text on soil type be shortened? F.e. "Unclear. All samples taken n temperate, oceanic locations"; Same row, last column: unclear what is meant with broad recommendation. What makes that a limitation?

L129: "suppress"

Table 3: Recommendation listed for Storage Methods is not formulated as a recommendation but reads like an observation. Replicates-Heterogeneity: authors recommend 5 replicates, but it is unclear on which this is based. Statistically, number of required replicates depends on the variation within the group/treatment.

L155: "in the literature are"

Supplementary material: Table S2 is strangely outlined in the text and stands separately from its caption.

Statistical analyses: how were normality and homogeneity assumptions checked?

L99: unclear which variables were log transformed before analysis. Is this natural logarithm or Log10?

Fig S6: see in general comments

---

## Referee Comment (RC2) · 23 Dec 2020

steven mcbride (Referee)

mcbridsg@vt.edu

The authors of this manuscript investigate the effects of storage method and duration on soil dissolved organic carbon (C) and nitrogen (N), and inorganic N. They identified common storage parameters through a literature survey and an online survey, and then tested how these common storage methods affected recovery of extractable dissolved C and N. Based on this experiment they report recommendations for storage of soil samples. This manuscript is important to Soil science community and fits well within the scope of SOIL. The manuscript highlights the of standardization in storage parameters when measuring dissolved C and N. Furthermore, they found that different

soils (subsoil vs topsoil) were affected differently by storage method. This manuscript was well written and pleasant to review. My major recommendations are related to the esentation of the case study data, and discussion of Soil storage methods. General comments: I believe the manuscript as a whole would benefit from including the major findings from the case study into the main text. These results are currently included in the Results section of the supplement. Furthremore, when included in the main text, the major findings of the case study should be briefly summarized in the abstract. Section 1 clearly states the importance of the standardizing soil storage methods.

Section 2 The Case Study is the strongest evidence that there is a problem with non standardized storage methods. Therefore, the results of the study should be included in the manuscript. The figures could remain supplementary since Table 2 summarizes the results. However, the Results should be moved to the main text with appropriate figure citations included in text. Line 93. Were the statistics performed in R? This should be stated and if the mixed effects models were performed using an R package that package should be cited. It appears they are cited in the supplement. That should be moved to the main text. Line 111-115 - Table 2: There appears to be a discontinuity between the table legend and the table itself. The table legend specifies that Red and Green squares, while the table has dark and light grey squares. It appears to be properly labeled in the note at the bottom of the table. Line 112-113 - I am not following the statement from the legend, "Green denotes appropriate storage method with additional recommendations for storage length". Where are the additional recommendations? If they are in the main text I believe that should be specified. Line 113 - I am also confused by this statement "Where we do not specify, stored samples did not differ from fresh samples through the entire experiment, 430 days." The only areas not specified are the dark grey boxes, which I understand to indicate an inappropriate storage method. Please clarify. Additionally, if they did not differ during that time, then the sign should be greater than or equal to, or simply stated that recommendations beyond 430 days can not be made due to the length of the experiment.

Section 3 and 4 are strong. However, it appears to be implied that these recommendations are specific to studies comparing different soil, e.g. topsoil vs subsoil. Based on the study and the recommendations, it does not appear to apply to researchers carrying out an ecological study using a single soil that has been experimentally manipulated. If the authors agree with my conclusion then they should state that their recommendations do not apply in the circumstance. If they do not, then I they should provide a strong argument for why not.

Section 5 The authors should also address experimental manipulations of single soil types here as well.

Technical notes Line 19 space needed between "andNO3" Line 31 I think this citation may be wrong in the citation manager because it does not have the author's name here or at the beginning of the citation in the references Line 60 Add a space after the 4 and before the degree symbol Table 1 Formatting to prevent last letter of some words moving to next line Line 180, I don't think the parentheses are necessary around 2020

---

## Short Comment (SC1) · 23 Dec 2020

Comment by Hanna Frick & Else K. Bünemann (with contributions from Maike Krauss, Andreas Fliessbach)

As a group of researchers from the Soil Science Department of the Research Institute of Organic Agriculture (FiBL), Switzerland, we discussed the manuscript entitled "Are researchers following best storage practices for measuring soil biochemical properties?" by Rhymes at al. 2020.

Rhymes et al. 2020 raise the discussion on an important topic that concerns the whole

soil science community. We would like to acknowledge the authors' enormous work in a comprehensive and valuable case study on best practice storage conditions for soil samples and soil extracts for various commonly investigated biochemical parameters (with almost 2000 extractions performed). We highly appreciate their initiative in raising awareness on this vital, but often neglected topic and hope that their contribution will spark further work and exchange among soil scientists.

However, in our opinion some important aspects were not considered adequately and we have the following suggestions for improvement:

1) The data on which Rhymes et al. base their guidance should be provided in the main manuscript rather than the supplementary information (SI). While the authors themselves claim that "[. . .] optimal storage conditions will vary across different soils and ecosystems" (Line 75), but also between top- and subsoil, as shown by their own case study, they come to very generalized recommendations on best practice storage in Table 2, which we find contradictory. To us it is not quite clear how the authors come to their recommendations, or at least some differentiation is lacking. For instance, in Figures S4a and S5a, storing frozen extracts up to 430 days seems tolerable for both MBC (only topsoil) and MBN (both top- and subsoil), but in Table 2 freezing extracts for assessment of microbial biomass is indicated as completely inappropriate.

2) The discussion of changes upon storage should be further elaborated and put in context with existing literature (e.g. the literature reviewed in Table 1). For example, Stenberg et al. (1998) suggest that soils can be stored frozen for up to 13 months for assessing microbial biomass, while Rhymes et al. recommend not to freeze soil at all for any kind of biochemical analysis they considered in their manuscript. How would the authors explain these differences? In Line 130, Rhymes et al. speculate about microbial processes as the main driver of changes in stored soils or extracts and they suggest storing samples under conditions which suppress microbial activity completely. Given the major changes still happening in frozen extracts over time for NO3 (Figure S3a), do the authors suggest that freezing is not suppressing microbial

activity sufficiently? Could there be other mechanisms responsible for this trend? The discussion currently provided in the SI should be moved to the main manuscript in order to increase its visibility to the scientific community.

3) The importance of the underlying research questions is neglected: The authors only look into relative changes in the measured parameters in comparison to freshly extracted and immediately analyzed samples. However, many studies aim at investigating relative differences between treatments rather than obtaining absolute data on fresh samples. In fact, appropriate storage conditions are not only part of the method, but also depend strongly on the research question. In many cases, standardized pre-treatments (for example pre-incubation of soil after refrigerated storage for microbial N and C), freezing of all samples before extraction etc. might produce smaller errors than immediate extractions, where differences upon sample collection, transport, outside temperature upon sampling etc. would arguably cause bigger effects than the storage treatment. With this regard, especially the change in the measured parameters upon prolonged duration of storage is relevant. For instance, Rhymes et al. consider freezing of soil or extract for analysis of NH4 inappropriate (Table 2 or Figure S3b), however, changes here seem to appear immediately upon freezing, with marginal changes thereafter (Figure S3b). For studies only interested in relative differences between treatments or sites, freezing thus would be a tolerable storage method. Again, we think that the recommendations should be more differentiated and take potential research questions into account.

4) From our own experience, but also highlighted by the results of the survey which Rhymes et al. conducted amongst different laboratories (note that the documentation on how the survey was conducted could be expanded), storage of both soil and soil extracts are common practice. This is owed to the mere impossibility to collect, extract and analyze samples in one day, especially with high sample numbers or when sample collection has to be conducted at large spatial distance to the lab. In this context, we find their conclusion on "appropriate" or "inappropriate" storage too general. How

about defining an acceptable relative error, e.g. by handling the samples in one way or the other? Furthermore, as indicated above, relative errors occurring immediately (e.g. upon freezing or un-freezing) should be distinguished from continued changes upon prolonged storage.

5) With their study, Rhymes et al. made an important point on the effect of storage conditions, but we miss the broader picture. The discussion should expand also on other aspects potentially compromising the integrity of soil samples, such as sampling procedure, transport, pre-treatments or handling of the samples in the laboratory. We believe that the whole soil science community should put more effort into defining common standards and evaluating potential errors during the whole procedure from sample collection, transportation and storage until analysis. Comparing the effect of storage conditions with the effects of these other aspects would help to identify sources of major errors and design experiments accordingly.

6) If each group has to carry out their own pilot studies and resulting storage conditions will vary substantially, then meta-analyses will become even more difficult than they are now. Besides, the recommendations for such pilot studies would need to be really concise, e.g. how many time points would need to be analyzed? It would be important to learn as much as possible from the experiment conducted by Rhymes et al. As an alternative to pilot studies, why not put an effort into identifying suitable reference materials that can be included in each study?

In addition to these general thoughts, here are some more detailed comments:

Sampling procedure and soil sample preparation

- While we understand the reasoning behind their sampling approach (topsoil sampled three weeks after sampling the subsoil), in most of our experiments this is simply not an option, e.g. due to distant sampling locations and the importance of a uniform sampling time point.

- Soil samples were taken in June. Would results be different if soils had been sampled in winter or at a different initial water content? Generally speaking, the effect of seasonality should be discussed.

- Apparently, Rhymes et al. use "field replicates" for their extractions (SI Line 43ff): Soil was sampled from five locations (transect over the field with 10m distance between plots) in 0.5 x 0.5 m pits. These replicates were later on used for the extraction/different storage treatments. This sampling approach explains the high data variability upon the individual time points and storage conditions and should have been discussed by the authors.

- The time of sieving/homogenization was not investigated, since Rhymes et al. sieved all soil samples on the day after sample collection and stored all the soils sieved. Would the results have been different if soils had been sieved only after storage, immediately before extraction?

Extraction procedure and handling of extracts

- SI Line 71: For $K_2SO_4$ extraction, no blanks were performed. While this seems valid for the calculation of microbial C and microbial N as difference between fumigated and non-fumigated extracts, we find this problematic for reporting values on total C and total N in both fumigated and non-fumigated extracts, which were not corrected for blanks (compare Figures S4 b, c and Figure S5 b, c)

- The molarities of extractants ($K_2SO_4$, KCl) are not reported throughout the whole manuscript.

- Scaling of extractions procedure: Authors report that 5 g of moist soil were extracted. This is a very low amount considering any potential inhomogeneity in the soil. Due to the high number of replicates (n=5) this might be acceptable. Additionally, soil moisture content (e.g. between top- and subsoil) was ignored upon extraction, which might lead to differences in the soil-to-solution ratio. Equal amounts of dry soil equivalents should

be used for a standardized extraction procedure.

- Scaling could also be added as another point to consider for a pilot study within Table 3 (extraction methods; recommendation: do not up-/down-scale the used amounts but use the same amounts as planned for the main experiment).

- Freezing and un-freezing procedures were not investigated as further factors. From our experience, it makes a difference in which position extracts are frozen (e.g. vertical or horizontal placement of tubes) and under which conditions extracts or soils samples are thawed (e.g. thawing soils over night at 4°C or extracting frozen soil immediately with the solution).

Statistics/Figures/Data presentation

- SI Line 104: Why did the authors use a plot digitizer to extract numeric data from their own plots?

- Figure S1b: In the figure caption, authors indicate that there was a technical problem with the DON measurement on the last time point (Day 430) and thus, data should not have been included. However, in the figure there a data points also for this sampling time.

- For some of the analyzed parameters, the replicates show a very high data variability. However, this seems not always represented in the confidence interval displayed (e.g. Figure S3 a: $NO_3$ values for frozen extracts vary widely, while the confidence interval seems to be very small).

- In Table 3, authors recommend to use twice the number of replicates for the baseline (freshly extracted and analyzed samples). However, for their own case study they did not follow this recommendation or at least did not report it.

Technical comments:

- Typo in Table 1: Plant available N, reference "Jones and Willett 2006", under storage

methods explored it should probably be -18°C

- Line 60: Wording is misleading. Stenberg et al. 1998 also sieved the soil prior to storing it at different temperatures.

- Table 2: There seems to be a mistake in the table header. We do not see any red or green squares. We assume that the information given below the table ("Dark grey denotes inappropriate storage method and light grey appropriate.") gives the same information?

- Line 135: Figure 1 should only have a figure caption below, but not additionally above.

- Table S2: Table header is missing.

STENBERG, B., JOHANSSON, M., PELL, M., SJÖDAHL-SVENSSON, K., STEN-STRÖM, J. & TORSTENSSON, L. 1998. Microbial biomass and activities in soil as affected by frozen and cold storage. Soil Biology and Biochemistry, 30, 393-402.

---

## Author Comment (AC1) · 5 Feb 2021

In this short communication the authors address the lack of evidence-based recommendations for storage of soil samples and extracts prior to analysis of organic and inorganic carbon (C) and nitrogen (N) and microbial biomass C and N. Based on literature and an online survey, the authors demonstrate that samples and extracts are stored in a multitude of ways. They further provide a case study in which they demonstrate the effects of sample and extract storage on measurements of organic, inorganic and microbial C and N, further stressing the need for standardisation of protocols.

The manuscript offers recommendations for sample and extract storage for the investigated methods, a flow-chart to guide researchers deciding on the most appropriate storage approach for their experiments and recommendations for reporting of the methods adopted for scientific publications. The results of the case study are a welcome reference for future research. In particular, the case study demonstrate that storage methods do not affect samples of different soil types (top vs subsoil) equally. This finding goes against the common assumption that storage affect all sample types similarly.

This short communication is relevant for the international science community, and fits well in the scope of SOIL. The introduction outlines the motivation of the objectives clearly, and the necessary details of the case study methodology and results are given in the supplementary material. The text is well written and pleasant to read. I have a few suggestions for corrections. Tables and figure are mostly supportive (particularly table 2). I have some suggestions to further clarify the text in tables 1 and 3. Also, I wonder whether the message the authors want to convey in section 4 with Table 4 isn't better served by integrating its information into the text. Lastly, I would ask the authors to consider including Fig S6 into the main text. I outline my general and minor comments in more detail below.

***Thank you very much for the positive comments and useful suggestions. We have detailed how we will address all the suggestions and recommendations proposed, which we believe will improve our manuscript. See details below.***

General comments: The authors conducted an online survey and reflect on the outcome in L65 and further. Adding few lines about the nature of the survey and how representative the responses are would increase the value of its outcome in the authors argumentation.

***As recommended, we will include information on the survey data collection. The survey was conducted anonymously on Google surveys and promoted through Twitter a social media platform. A total of 68 participants provided information on how they typically store their soil and/or extract samples. Unfortunately, we do not have any survey participant demographics (e.g. types of institution, geography, positions of those who answered). Although this would have provided valuable insight as to who carries out specific storage methods and the representativeness of the responses, we decided that responses would remain anonymous as we felt that some researchers may not want to disclose how long samples may be left in storage prior to analysis.***

***We will add the survey questions below to supplementary materials as detailed below:***

1) ***Realistically, how do you process your samples prior to carbon and nitrogen analysis (KCl, $K_2SO_4$, $H_2O$)?***
    i) ***Extract and run all my samples immediately***
    ii) ***Extract immediately and store the extract until analysis***
    iii) ***Store the soil and carry out extraction for immediate analysis***
    iv) ***Store the soil and store the extract***
2) ***At what temperature do you generally store your extracts?***
    i) ***4°C***

      *ii) -20°C*

      *iii) -80°C*

      *iv) Room temperature*

      *v) I do not store extracts prior to analysis*

      *vi) Other:*

*3) At what temperature do you generally store your soil?*

      *i) 4°C*

      *ii) -20°C*

      *iii) -80°C*

      *iv) Room temperature*

      *v) I do not store soil prior to analysis*

      *vi) Other:*

*4) How long do you typically store your soils for prior to extraction?*

      *i) 2 days or less*

      *ii) 2 days - 1 week*

      *iii) 1 - 3 weeks*

      *iv) 3 weeks - 2 months*

      *v) Longer than 2 months*

      *vi) I do not store soil prior to analysis*

*5) How long do you typically store your extracts for prior to analysis?*

      *i) 2 days or less*

      *ii) 2 days - 1 week*

      *iii) 1 - 3 weeks*

      *iv) 3 weeks - 2 months*

      *v) Longer than 2 months*

      *vi) I do not store extracts prior to analysis*

*6) Any additional comments?*

Also, inclusion of Figure S6 into the introduction would aid the reader follow the narrative, or at least place it first in the Supplementary materials. For Figure S6, I recommend the authors to clarify the meaning of the colours, explain the abbreviation of RT, and consider scaling the size of the arrows to the % of cases.

*We agree that we should move Fig S6 to the start of the supplementary material. We will also clarify in the figure caption what the colours mean and scale the arrows, as requested; please see the new figure below. The new figure caption will read: "Flow chart indicating the results of an online survey to determine how and for how long people store soils and/or extracts prior to analysis. Room temperature is abbreviated as RT. Survey date: December 2018 – June 2019. The proportion of people that followed each methodological step is indicated as 'n' in each arrow, which are also scaled in size depending on 'n'. The arrow colours correspond to the differences in soil sample storage, processing and extract storage prior to analysis. Pink: soil samples are extracted immediately after collection, extracts are stored for later analysis. Blue: Neither soils nor extracts are stored. Green: soils are stored after collection, but extracts are analysed immediately after extraction. Yellow: both soils and extracts are stored."*

[Figure]

I miss the mentioning of other storage methods than those considered by the authors in their case study. L73 states the authors considered widely used storage methods, but it does not become clear to me how the selected methods compare to alternatives, for example drying. I recognize that a full evaluation is beyond the scope of this short communication yet adding a few examples could bring the proposed manuscript into a broader perspective. For example, storage and the achievability of soil samples are important considerations when choosing indicators for monitoring efforts (Ritz K, Black HIJ, Campbell CD, et al (2009) Ecol Indic 9:1212–1221. https://doi.org/10.1016/j.ecolind.2009.02.009).

*In response to the reviewer's comment, we will also expand on other storage alternatives in the introduction section to broaden the perspective of this short communication. We will add the following text to line 33 "In these cases, it is common practice to store samples for future analysis of which there are a broad range of storage practices. These include freeze drying, air drying, freezing and refrigerating samples and are typically chosen dependant on the analysis in question and time in which analysis can take place."*

*We will also add a short discussion as to why we chose to explore refrigeration and freezing treatments. From line 69, we will add the following text: "In our case study we chose to explore refrigerating and freezing storage practices in comparison to other storage methods (e.g. air drying or freeze drying) because there is significant evidence to suggest that those methods are unsuitable for the variables we measure. For example, air drying soils has a strong effect on C and N pools, probably due to microbial death and nutrient release upon drying and rewetting (Jones and Willett, 2006; Kaiser et al., 2001; Li et al., 2012; Rolston and Liss, 1989). Additionally, freeze-drying is also known to have a strong effect on nutrient pools, as the chemical, physical, and physiological stresses inflicted by freeze-drying severely injure or kill most soil microbes, releasing the microbial compounds into the soil (Islam et al., 1997).*

*We would also like to highlight that our chosen storage treatments are commonly practiced as highlighted by the survey: 66% of participants store their soils at 4°C or -20°C whilst 87% of participants store their extracts at 4°C or -20°C.*

While sections 2 and 3 give helpful directions to soil scientists, I find section 4 less strong. Table 4 presents best reporting practices, yet the section does not mention why the listed requirements are important, nor gives a motivation for providing these recommendations. Also, I would recommend the authors to remove table 4, and instead list the four reporting recommendations in the text in section 4. To convey the best reporting practice, I don't think it is necessary to provide examples of poor reporting and giving examples of good reporting are easier to read when listed in the text.

***We agree and will remove table 4 and integrate some of the information in the table into the text. We will also highlight why it is so important to report storage methods in section 4. This will include data that can later be used for meta-analysis.***

Minor issues:

L43: "that have considered these have taken into account few…" makes it easier to read

***This will be made clearer: "In many comparative studies exploring the impacts from methodological factors overlook soil and/or extract storage temperatures and duration, and those that have considered these, have taken into account few storage possibilities (Table 1)".***

Table 1: Column 6 "storage methods explores", remove the word 'only' in the listed examples.

***We will remove only throughout the table.***

Table 1: Yolo loam: what soil type reference was used, e.g. WRB soil types?

***These are Typic Xerorthents, USDA classification.*** https://soilseries.sc.egov.usda.gov/OSD_Docs/Y/YOLO.html

https://www.nrcs.usda.gov/Internet/FSE_DOCUMENTS/nrcs142p2_051232.pdf

Table 1: $H_2O$ – Rolson and Liss: at what temperatures were the soils stored frozen?

***We will amend this and add -10 °C***

Table 1: Plant available N – Soil Type: what is the difference between not applicable and not provided?

***We use the term "not applicable" for general recommendations that are made in the literature but are not based on comparative studies or investigations. We use the term "not provided" for comparative studies that do not describe the soils explored for different storage methods. We will make this clear in the table caption, which will be revised to read: "Summary of different recommendations for storage of soil or extract samples to measure soil nutrients found in the literature. This summary is non-exhaustive. The term not applicable under soil type refers to studies that were not based on comparative studies and therefore were not carried out on a soil type. The term not provided for refers to comparative studies that do not describe the soils explored in the methods"***

Table 1: Plant available N – Jones & Willett 2006: can the text on soil type be shortened? F.e. "Unclear. All samples taken from temperate, oceanic locations";

***We are happy to make this change.***

Table 1: Same row, last column: unclear what is meant with broad recommendation. What makes that a limitation?

***We will make this clearer. The recommendation made in the referenced publication states "store the extract for days". It is unclear how many days. We will change this to "Vague recommendations made for extract storage length which could be open to different interpretation".***

L129: "suppress"

*We will change this to "halt" as suggested by reviewer S1.*

Table 3: Recommendation listed for Storage Methods is not formulated as a recommendation but reads like an observation.

*We agree with this point. As we will include more details on alternative storage methods in the introduction, we will remove this row from the Table 3.*

Table 3: Replicates-Heterogeneity: authors recommend 5 replicates, but it is unclear on which this is based. Statistically, number of required replicates depends on the variation within the group/treatment.

*We take this onboard and would like to address this issue by amending our recommendation in Table 3 to read: "Generally, we recommend as many replicates as one can afford to have, but recommend no fewer than 4 as suggested by Jones and Willet (2006). For more guidance on choosing the number of replicates, we advise researchers to utilise the sample size calculator formula from Cochran and Cox (1957), p. 20. ".*

L155: "in the literature are"

*This will be corrected.*

Supplementary material: Table S2 is strangely outlined in the text and stands separately from its caption.

*We will ensure the position of the table caption is above the table*

Statistical analyses: how were normality and homogeneity assumptions checked?

*We will include this information to our statistical analysis. "Normality and homoscedasticity of the data were first checked using Anderson Darling and Levene's tests respectively."*

L99: unclear which variables were log transformed before analysis. Is this natural logarithm or Log10?

*We will include this information in the statistical analysis section of the manuscript. "All variables were subjected to natural log transformation except for DOC which was not transformed."*

Fig S6: see in general comments

*We will move Fig S6 to the start of the supplementary.*

**References**

Cochran, W.G., Cox, G.M., 1957. Experimental designs. john willey and sons. Inc., New York 546–568.

Islam, K.R., Weil, R.R., Mulchi, C.L., Glenn, S.D., 1997. Freeze-dried soil extraction method for the measurement of microbial biomass C. Biol. Fertil. Soils 24, 205–210. https://doi.org/10.1007/s003740050232

Jones, D.L., Willett, V.B., 2006. Experimental evaluation of methods to quantify dissolved organic nitrogen (DON) and dissolved organic carbon (DOC) in soil. Soil Biol. Biochem. 38, 991–999. https://doi.org/10.1016/j.soilbio.2005.08.012

Kaiser, K., Kaupenjohann, M., Zech, W., 2001. Sorption of dissolved organic carbon in soils: effects of soil sample storage, soil-to-solution ratio, and temperature. Geoderma 99, 317–328.

https://doi.org/https://doi.org/10.1016/S0016-7061(00)00077-X

Li, K. yi, Zhao, Y. yuan, Yuan, X. long, Zhao, H. bing, Wang, Z. hui, Li, S. xiu, Malhi, S.S., 2012. Comparison of Factors Affecting Soil Nitrate Nitrogen and Ammonium Nitrogen Extraction. Commun. Soil Sci. Plant Anal. 43, 571–588. https://doi.org/10.1080/00103624.2012.639108

Rolston, D.E., Liss, H.J., 1989. Spatial and temporal variability of water-soluble organic carbon in a cropped field. Hilgardia 57, 1–19. https://doi.org/10.3733/hilg.v57n03p019

---

## Author Comment (AC2) · 5 Feb 2021

Thank you very much for your response. We have addressed all of your recommendations, which we believe will strengthen the manuscript. We provide a detailed response to each recommendation in the attached PDF.

Please also note the supplement to this comment:
https://soil.copernicus.org/preprints/soil-2020-79/soil-2020-79-AC2-supplement.pdf

---

## Author Comment (AC3) · 5 Feb 2021

As a group of researchers from the Soil Science Department of the Research Institute of Organic Agriculture (FiBL), Switzerland, we discussed the manuscript entitled "Are researchers following best storage practices for measuring soil biochemical properties?" by Rhymes at al. 2020. Rhymes et al. 2020 raise the discussion on an important topic that concerns the whole soil science community. We would like to acknowledge the authors enormous work in a comprehensive and valuable case study on best practice storage conditions for soil samples and soil extracts for various commonly investigated biochemical parameters (with almost 2000 extractions performed). We highly appreciate their initiative in raising awareness on this vital, but often neglected topic and hope that their contribution will spark further work and exchange among soil scientists.

**Thank you for the positive response. We appreciate the evaluation your group have made and we respond to the comments individually.**

However, in our opinion some important aspects were not considered adequately and we have the following suggestions for improvement:

1) The data on which Rhymes et al. base their guidance should be provided in the main manuscript rather than the supplementary information (SI). While the authors themselves claim that "[. . .] optimal storage conditions will vary across different soils and ecosystems" (Line 75), but also between top- and subsoil, as shown by their own case study, they come to very generalized recommendations on best practice storage in Table 2, which we find contradictory. To us it is not quite clear how the authors come to their recommendations, or at least some differentiation is lacking. For instance, in Figures S4a and S5a, storing frozen extracts up to 430 days seems tolerable for both MBC (only topsoil) and MBN (both top- and subsoil), but in Table 2 freezing extracts for assessment of microbial biomass is indicated as completely inappropriate.

As recommended, we will expand Section 2 by moving the entire results section from the supplementary into the main manuscript, which will include line 107 to line 172 from the supplementary. The figures will remain in the supplementary and will be referred to in the main manuscript.

Summarising our findings into generalised recommendations was a particularly difficult task considering the differences we found between soil depths. We also want to highlight that these are not guidelines that should be followed if using different soil types or soils at different depths to those explored. The general rational for the guidelines summarised in Table 2 were that: 1) a storage method must have deemed appropriate for both subsoils and topsoils and 2) where the same extractant is used to measure different parameters they too must deem appropriate for both sets of parameters. In the example you give for MBC and MBN, we would like to highlight that storing frozen extracts was not deemed appropriate for measuring MBC in subsoils and therefore did not meet our first clause. Furthermore, microbial biomass is generally measured as microbial biomass nitrogen and carbon and would use the same extraction solution to measure both and consequently does not meet our second clause.

To clarify this issue, we will add this statement to line 230: "However, freezing samples did not significantly affect the concentration of N in fumigated or unfumigated samples, and thus frozen extracts was a suitable storage method to measure MBN."

We will also clarify our rational for making such guidelines in the table caption. This will be revised to read: "Table 2. Storage method recommendations for both temperate topsoil and subsoil. Dark grey denotes inappropriate storage methods for a specific analysis. Light grey denotes appropriate storage method, where storage length is annotated. Where storage length is annotated as 430 days we are unable to advise storage length beyond this due to the length of the experiment. Storage methods are deemed appropriate: 1) if the storage method does not compromise the sample integrity (defined as stored samples yielding soil parameter values within 20% similarity limits to fresh samples) for both topsoil and subsoils explored; and 2) where the same extractant type is used to measure different parameters, the storage method does not compromise the integrity of each parameter measured."

We do also provide some guidance on this issue in Table 3 bottom row "deciding on the best storage practice" but hope that with the added information in the caption this will be much clearer.

2) The discussion of changes upon storage should be further elaborated and put in context with existing literature (e.g. the literature reviewed in Table 1). For example, Stenberg et al. (1998) suggest that soils can be stored frozen for up to 13 months for assessing microbial biomass, while Rhymes et al. recommend not to freeze soil at all for any kind of biochemical analysis they considered in their manuscript. How would the authors explain these differences?

We will expand the discussion and compare our results with those published in the literature (see below for more details). For example, the key difference between our study and Stenberg et al. is that they only measured MBC, whilst we measured both MBC and MBN. In line with the findings of Stenberg et al. we also found that freezing soils to measure MBC was acceptable (up to 75 days for topsoil and 430 for subsoil), but not for MBN (although acceptable for topsoil up to 430 days). As discussed above, due to subsoil sample integrity being jeopardised by freezing soils to quantify MBN we deemed this storage method inappropriate.

Another potential explanation for these differences could be differences in soil microbial communities. They worked with soils from Upsala, Sweden that experience marginally lower average winter temperatures than the soils we collected. In turn, the microbial communities in their soils could be more adapted to colder temperatures causing them to respond differently to freezing storage methods.

In Line 130, Rhymes et al. speculate about microbial processes as the main driver of changes in stored soils or extracts and they suggest storing samples under conditions which suppress microbial activity completely. Given the major changes still happening in frozen extracts over time for NO3 (Figure S3a), do the authors suggest that freezing is not suppressing microbial activity sufficiently? Could there be other mechanisms responsible for this trend?

We do not explore the mechanisms responsible for the changes to samples under different storage conditions and therefore can only discuss potential mechanisms. Indeed, freezing does suppress activity, but our results demonstrate that this is not enough to maintain sample integrity. As a result, (on line 130) we recommend other potential storage options that completely halt microbial activity such as the acidification of extractants. To avoid confusion, we will change "completely suppress microbial activity" to "completely halt microbial activity".

In response to freezing not supressing microbial activity, there is evidence for this; freezing has been shown to decrease microbial biomass as a result of damaged cell structures (Černohlávková et al., 2009). As a result microbes that are not damaged by the freezing process profit from the organic molecules made available from the proportion of microbes that have died (Stenberg et al., 1998). This could be a potential mechanism to explain the shift over time whilst in the freezer but we are unable to conclude this as we did not investigate the mechanisms.

The discussion currently provided in the SI should be moved to the main manuscript in order to increase its visibility to the scientific community.

We agree with the reviewer's comments and would like to include some of the potential mechanisms for the sample deterioration we observed under different storage conditions. We will move paragraphs starting from line 182 to 230 from the supplementary material into a new section with two subsections to discuss mechanisms involved in storing soils and extracts (Section 2.3 – Results and discussion).

3) The importance of the underlying research questions is neglected: The authors only look into relative changes in the measured parameters in comparison to freshly extracted and immediately analyzed samples. However, many studies aim at investigating relative differences between treatments rather than obtaining absolute data on fresh samples. In fact, appropriate storage conditions are not only part of the method, but also depend strongly on the research question. In many cases, standardized pretreatments (for example pre-incubation of soil after refrigerated storage for microbial N and C), freezing of all samples before extraction etc. might produce smaller errors than immediate extractions, where differences upon sample collection, transport, outside temperature upon sampling etc. would arguably cause bigger effects than the storage treatment. With this regard, especially the change in the measured parameters upon prolonged duration of storage is relevant. For instance, Rhymes et al. consider freezing of soil or extract for analysis of NH4 inappropriate (Table 2 or Figure S3b), however, changes here seem to appear immediately upon freezing, with marginal changes thereafter (Figure S3b). For studies only interested in relative differences between treatments or sites, freezing thus would be a tolerable storage method. Again, we think that the recommendations should be more differentiated and take potential research questions into account.

We agree that other procedures for sample collection and preparation will also affect sample integrity. We are happy to refer to these other procedures in the introduction to ensure readers also consider other aspects that will impact sample integrity. This will include sample collection, transport, processing and analytical practices that can also influence results. We will add these sentences to the introduction from Line 30 to address this issue: "It is therefore integral that researchers consider each factor that can impact accurate and reliable analytical measurements, which can include sampling procedures (e.g. strip removal of turf), transport (e.g. transport length and temperature), storage (e.g. temperature), preparation for analysis (e.g. sieving mesh size and when samples are sieved) and analytical methods (e.g. temperature, shaking times and filter types). Here we focus solely on sample storage. While most soil biogeochemical analyses should ideally be carried out on fresh samples immediately after sampling (ISO18400-102:2017, 2017), this is not always possible due to the number of samples taken and the analytical procedures exceeding human and/or instrumental capabilities. In these cases, it is common practice to store samples for future analysis."

However, we disagree with the reviewers' view that freezing is a tolerable method for storage when studies are only looking at relative differences between treatments and/or site. Our findings demonstrate that storage can impact soil from different depths differently, which implies that soils of different types are likely to respond differently to storage methods. We did not explore the mechanisms for the differences observed and therefore cannot assume that this is correlated to soil depth alone. It might be as a result of other factors such as differences in nutrient status or microbial populations for example. If this were to be true, two samples of the same soil type with different nutrient statuses would respond differently to a soil storage method. Or two soils under different experimental treatments that had affected their nutrient status or microbial communities would also respond differently. We therefore do not recommend that researchers assume that samples from the same site will respond the same to a storage method and that the relative differences between treatments will not change. Additionally, not all variables respond in the same way to storage, and if researchers are to study C/N ratios for example, storage effect will have a strong impact on them. Finally, it is true that there are some changes that occur immediately and some that occur over storage time. However, this immediate change was not equal in the two soil depths studied (for example frozen extracts for NH4 determination, or freezing soil for DON), and hence, the same logic applies.

We will now include this in our discussion to address this issue: "It is commonly assumed that any changes to soil biochemistry from storage methods will occur equally for all samples. Here, we provide evidence to show that changes do not occur equally which could have major implications for the findings of ecological studies. We did not investigate the mechanism behind different responses of two soils to the storage treatments, but it could be related to many factors such as differences in nutrient status or microbial communities. As a result, any treatment that affects soil properties have the potential to also affect the response of soils to storage. Even if sample biochemistry changes immediately as a result of storage but subsequently remains stable over storage time, in our study this effect varied between the two soil depths. Therefore, even if the research question is to compare between treatments applied to the same soil type, strict storage limits should be followed. We suggest that all samples should be stored under the same conditions that allow the preservation of samples from the soil type, site and/or treatment with the highest sensitivity to storage."

4) From our own experience, but also highlighted by the results of the survey which Rhymes et al. conducted amongst different laboratories (note that the documentation on how the survey was conducted could be expanded), storage of both soil and soil extracts are common practice. This is owed to the mere impossibility to collect, extract and analyze samples in one day, especially with high sample numbers or when sample collection has to be conducted at large spatial distance to the lab. In this context, we find their conclusion on "appropriate" or "inappropriate" storage too general. How about defining an acceptable relative error, e.g. by handling the samples in one way or the other? Furthermore, as indicated above, relative errors occurring immediately (e.g. upon freezing) should be distinguished from continued changes upon prolonged storage.

We agree that we need to provide more information about our survey approach. The survey was conducted on Google surveys and promoted through twitter a social media platform. We will now include the survey questions in supplementary materials (see details in response to referee 1). We will now include the rational for the survey and the survey questions.

Regarding the "appropriate" or "not appropriate" recommendations, they are already taking into account and accepting a 20% error for each storage treatment explored. But if researchers consider that a higher % error is acceptable for a particular study due to strong logistical limitations, a higher error to determine appropriateness could be used. Regarding changes happening immediately upon storage, please, refer to our response above.

5) With their study, Rhymes et al. made an important point on the effect of storage conditions, but we miss the broader picture. The discussion should expand also on other aspects potentially compromising the integrity of soil samples, such as sampling procedure, transport, pre-treatments or handling of the samples in the laboratory. We believe that the whole soil science community

should put more effort into defining common standards and evaluating potential errors during the whole procedure from sample collection, transportation and storage until analysis. Comparing the effect of storage conditions with the effects of these other aspects would help to identify sources of major errors and design experiments accordingly.

We agree that the manuscript would be strengthened if a comment were to be added to the introduction to address other aspects of sample collection and processing beyond storage methods can also impact sample integrity. We have addressed this in your 3rd comment (please see 3 above) and will also include an amended sentence in section 5: Conclusions.

"We stress that researchers must also consider other practices beyond just storage (e.g. sieving samples, transport, extraction procedures...) as each methodological step between sample collection and analysis can introduce errors to measurements that are intended to be field representative. We encourage researchers to utilise standardised methods where possible (see e.g. Halbritter et al. (2020)) and to follow best storage practices for specific soil types to allow reliable comparison of data from different studies."

6) If each group has to carry out their own pilot studies and resulting storage conditions will vary substantially, then meta-analyses will become even more difficult than they are now. Besides, the recommendations for such pilot studies would need to be really concise, e.g. how many time points would need to be analyzed? It would be important to learn as much as possible from the experiment conducted by Rhymes et al. As an alternative to pilot studies, why not put an effort into identifying suitable reference materials that can be included in each study?

We believe that if more people that publish and report their pilot studies, this will generate the data required to conduct a large meta-analysis that paints a much clearer picture as to how storage impacts our results. This could potentially result in a standardised storage protocol being produced. But to date we know very little about the impacts of storage methods. In the supplementary material "Extended material and methods", we provide all the details necessary to replicate our study. Furthermore, in Table 3, we consider the different aspects that researchers should take into account when designing their own pilot studies. We do not suggest or think that it should be necessary that all the pilot studies should be an exact replica of our study. It is more appropriate that they focus on the aspects (e.g. storage methods, length, similarity limit) that can better inform their own experiments. Indeed, storage conditions will vary amongst these pilot studies but high variability in ecological studies already exist too and are commonly used for meta-analysis. Efforts should still be made despite this hurdle. We believe this commentary will spark the necessary discussions amongst the soil community to give sample storage more consideration than it currently does.

Suitable reference materials may be an option for quality control purposes. However, this would need to be explored for future recommendation as little is known about the mechanisms behind the shifts associated with storage and most importantly whether reference materials would behave similarly to living soils.

In addition to these general thoughts, here are some more detailed comments:

Sampling procedure and soil sample preparation

- While we understand the reasoning behind their sampling approach (topsoil sampled three weeks after sampling the subsoil), in most of our experiments this is simply not an option, e.g. due to distant sampling locations and the importance of a uniform sampling time point.

**This approach was due to logistical constraints, and we do not suggest that researchers stagger their sampling times for different soils routinely.**

- Soil samples were taken in June. Would results be different if soils had been sampled in winter or at a different initial water content? Generally speaking, the effect of seasonality should be discussed.

We agree that seasonality will affect any soil biochemical results measured. It is unknown whether they would also respond differently to storage methods.

We appreciate we do not make note of this in the manuscript and agree that we should. We will add this sentence in the discussion to address this comment: "We would like to note that due to the high temporal variability that the temperate soils explored experience, there is the potential that storage methods could impact sample integrity differently depending on when the samples were collected. Understanding the mechanisms responsible for jeopardising sample integrity under different storage methods will help determine the best storage methods for the time in which samples are collected (e.g. season), soil type and depth."

- Apparently, Rhymes et al. use "field replicates" for their extractions (SI Line 43ff): Soil was sampled from five locations (transect over the field with 10m distance between plots) in 0.5 x 0.5 m pits. These replicates were later on used for the extraction/different storage treatments. This sampling approach explains the high data variability upon the individual time points and storage conditions and should have been discussed by the authors.

We agree that true field replicates always result in higher variability, but we used this approach to avoid pseudo replication, and to provide a more robust representation of variability within our field experiment. We calculated relative change to help account for high variability. Additionally, we included a 20% similarity limit, instead of a stricter 10% similarity limit, which also helps accounting for this high variability.

We will include a statement on this issue in the supplementary methods in line 44: "This approach of sampling and keeping separate true field replicates was chosen to avoid pseudo replication, and to properly represent the high variability associated with typical ecological field experiments. High data variability was accounted for by calculating relative change for each individual replicate compared to corresponding fresh samples, and by increasing the acceptable similarity limit to 20% (see statistical analyses for details)."

We consider that adding that information in the methods is sufficient and we do not feel it is necessary to include a statement in the discussion.

- The time of sieving/homogenization was not investigated, since Rhymes et al. sieved all soil samples on the day after sample collection and stored all the soils sieved. Would the results have been different if soils had been sieved only after storage, immediately before extraction? Extraction procedure and handling of extracts

Yes, this will certainly yield different results, we address this comment both in the introduction and conclusion (Please see our response to points 3 and 5). We do not feel that it is necessary to discuss in too much detail as the purpose of this study was to look at storage methods. We feel that with the inclusion of the sentences we propose in the introduction and the conclusion we have made it much clearer that researchers need to consider all of these aspects including sieving, from sieving mesh size to when the soils are sieved immediately after collection or just before analysis.

- SI Line 71: For K2SO4 extraction, no blanks were performed. While this seems valid for the calculation of microbial C and microbial N as difference between fumigated and non-fumigated extracts, we find this problematic for reporting values on total C and total N in both fumigated and non-fumigated extracts, which were not corrected for blanks (compare Figures S4 b, c and Figure S5 b, c)

We agree that our results would have benefit from K2SO4 blanks. When the experiment was designed, we did not consider the possibility of presenting the raw unfumigated and fumigated data, and only the microbial C and N, deeming unnecessary to process and store K2SO4 blanks (which would have increase the number of extracts to an additional 180 samples). However, for the purpose of this study, we are only looking at relative differences between storage treatments and feel that presenting the raw unfumigated and fumigated data without blank correction is acceptable. In accordance, the recommendations we make are based on the microbial C and N values and not the unfumigated and fumigated values. The fumigated and unfumigated samples only helped in the discussion to explain some of the effects observed.

- The molarities of extractants (K2SO4, KCI) are not reported throughout the whole manuscript.

**This will be added throughout. The molarity used was 0.5M for K2SO4 and 1M for KCl.**

- Scaling of extractions procedure: Authors report that 5 g of moist soil were extracted. This is a very low amount considering any potential inhomogeneity in the soil. Due to the high number of replicates (n=5) this might be acceptable.

**This is very common practice and is reflected in numerous publications, some even use less soil (e.g. 2.5g of soil to 25ml extractant in Jones and Willett).**

Additionally, soil moisture content (e.g. between top- and subsoil) was ignored upon extraction, which might lead to differences in the soil-to-solution ratio. Equal amounts of dry soil equivalents should be used for a standardized extraction procedure.

Although we did not utilise equal amounts of dry soil equivalents, we did correct for differences in soil weights as a result of differences in soil moisture for final nutrient concentration calculations. It has come to our attention that this is not appropriately detailed in our methods and we will amend this. We would like to highlight that it is common practice to correct for the difference in dry weight when calculating concentrations, please see the standardised protocol in Halbritter et al. (2020). We would also like to draw your attention to Table S1 and note that we only saw a 2% variation in soil moisture and therefore feel that this will not have impacted our results.

- Scaling could also be added as another point to consider for a pilot study within Table 3 (extraction methods; recommendation: do not up-/down-scale the used amounts but use the same amounts as planned for the main experiment).

We already elude to this concept in table 3, under replicates, pseudoreplication "Do not store soils or extracts in bulk. The same weight or volume of soil or extract must be stored separately for each storage treatment and time point." But we agree that it could include the idea of scaling down. We will modify this and create another row in the table for scaling. It will read:

"Consideration: Scaling Issues: pseudoreplication, reproducibility. Recommendations: Do not scale your soils or extracts for storage up (bulk storage) or down. The same weight or volume of soil or extract must be stored separately for each storage treatment and time point as the one planned for the main experiment." - Freezing and un-freezing procedures were not investigated as further factors. From our experience, it makes a difference in which position extracts are frozen (e.g. vertical or horizontal placement of tubes) and under which conditions extracts or soils samples are thawed (e.g. thawing soils over night at  $4\circ$ C or extracting frozen soil immediately with the solution).

This is an interesting point and would make for a good exploratory study. We will add details on the specific conditions in which extracts were frozen in the methods (they were all frozen vertically). Details on how soils and extracts were thawed are already in the supplementary information, extended methods (please see line 56).

We will add the idea of the potential effects of freeing/thaw procedures on measurements in the discussion. We will add this sentence: "Due to the potential for freeze-thaw cycles to impact sample biogeochemistry (Černohlávková et al., 2009) it is important to consider and be consistent with the freeze/thaw procedure, such as the position in which extracts are frozen (vertical or horizontal placement of tubes) or under which conditions extracts or soils samples are thawed (e.g. thawing soils over night at 4°C or extracting frozen soil immediately with the solution)."

Statistics/Figures/Data presentation

SI Line 104: Why did the authors use a plot digitizer to extract numeric data from their own plots?

We define sample integrity as unusable when the confidence interval of the predicted model intercepts the outline of the similarity limit which is shaded in grey. We didn't consider it was important as to how we calculated this and chose what seemed to be the easiest way. This can of course be calculated through the predicted confidence intervals from our models, which has yielded the same results. We will amend this accordingly in the methods.

- Figure S1b: In the figure caption, authors indicate that there was a technical problem with the DON measurement on the last time point (Day 430) and thus, data should not have been included. However, in the figure there a data points also for this sampling time.

Graph S1b is correct. However, you have brought to our attention that the 430 day sampling point was not visible on all other graphs due to the x axis only going up to 6 rather than 6.06 (log of 430 days). This is a graphical plotting error rather than a statistical error, whereby no sampling points were excluded from out the linear mixed models. We will amend all the graphs. As an example, here is our corrected graph forS4a, which now includes the last time point.

- For some of the analyzed parameters, the replicates show a very high data variability. However, this seems not always represented in the confidence interval displayed (e.g. Figure S3 a: NO3 values for frozen extracts vary widely, while the confidence interval seems to be very small).

**We would like to highlight that these are 95% upper and lower confidence intervals for the predictive fitted ratio change values based on the mixed effects models carried out. This is not an individual confidence interval at each time point, and that is the reason in this instance why data variability is high, but the represented confidence interval is small.**

- In Table 3, authors recommend to use twice the number of replicates for the baseline (freshly extracted and analyzed samples). However, for their own case study they did not follow this recommendation or at least did not report it.

**We did not use twice the number of replicates, but upon reflection for such a study we think this is valuable and have therefore added it into the recommendations.**

Technical comments:

- Typo in Table 1: Plant available N, reference "Jones and Willett 2006", under storage methods explored it should probably be -18 °C

**This shall be corrected.**

- Line 60: Wording is misleading. Stenberg et al. 1998 also sieved the soil prior to storing it at different temperatures.

**We have removed the word sieved so it is no longer misleading.**

- Table 2: There seems to be a mistake in the table header. We do not see any red or green squares. We assume that the information given below the table ("Dark grey denotes inappropriate storage method and light grey appropriate.") gives the same information?

**We will correct this and add grey colour information to the table caption. This will read:**

"Table 2. Storage method recommendations for both temperate topsoil and subsoil. Dark grey denotes inappropriate storage methods for specific analysis. Light grey denotes appropriate storage method, where appropriate storage length is annotated. Where storage length is annotated as 430 days we are unable to advise storage length beyond this due to the length of the experiment."

- Line 135: Figure 1 should only have a figure caption below, but not additionally above.

**This shall be corrected.**

- Table S2: Table header is missing.

**This shall be corrected. Table S2 header: "Table S2 Summary of extraction methods used in this experiment"**

**References**

Černohlávková, J., Jarkovský, J., Nešporová, M., Hofman, J., 2009. Variability of soil microbial properties: Effects of sampling, handling and storage. Ecotoxicol. Environ. Saf. 72, 2102–2108. https://doi.org/10.1016/j.ecoenv.2009.04.023

Halbritter, A.H., De Boeck, H.J., Eycott, A.E., Reinsch, S., Robinson, D.A., Vicca, S., Berauer, B.,

Christiansen, C.T., Estiarte, M., Grünzweig, J.M., Gya, R., Hansen, K., Jentsch, A., Lee, H., Linder, S., Marshall, J., Peñuelas, J., Kappel Schmidt, I., Stuart-Haëntjens, E., Wilfahrt, P., Vandvik, V., Abrantes, N., Almagro, M., Althuizen, I.H.J., Barrio, I.C., Te Beest, M., Beier, C., Beil, I., Carter Berry, Z., Birkemoe, T., Bjerke, J.W., Blonder, B., Blume-Werry, G., Bohrer, G., Campos, I., Cernusak, L.A., Chojnicki, B.H., Cosby, B.J., Dickman, L.T., Djukic, I., Filella, I., Fuchslueger, L., Gargallo-Garriga, A., Gillespie, M.A.K., Goldsmith, G.R., Gough, C., Halliday, F.W., Hegland, S.J., Hoch, G., Holub, P., Jaroszynska, F., Johnson, D.M., Jones, S.B., Kardol, P., Keizer, J.J., Klem, K., Konestabo, H.S., Kreyling, J., Kröel-Dulay, G., Landhäusser, S.M., Larsen, K.S., Leblans, N., Lebron, I., Lehmann, M.M., Lembrechts, J.J., Lenz, A., Linstädter, A., Llusià, J., Macias-Fauria, M., Malyshev, A. V., Mänd, P., Marshall, M., Matheny, A.M., McDowell, N., Meier, I.C., Meinzer, F.C., Michaletz, S.T., Miller, M.L., Muffler, L., Oravec, M., Ostonen, I., Porcar-Castell, A., Preece, C., Prentice, I.C., Radujković, D., Ravolainen, V., Ribbons, R., Ruppert, J.C., Sack, L., Sardans, J., Schindlbacher, A., Scoffoni, C., Sigurdsson, B.D., Smart, S., Smith, S.W., Soper, F., Speed, J.D.M., Sverdrup-Thygeson, A., Sydenham, M.A.K., Taghizadeh-Toosi, A., Telford, R.J., Tielbörger, K., Töpper, J.P., Urban, O., van der Ploeg, M., Van Langenhove, L., Večeřová, K., Ven, A., Verbruggen, E., Vik, U., Weigel, R., Wohlgemuth, T., Wood, L.K., Zinnert, J., Zurba, K., 2020. The handbook for standardized field and laboratory measurements in terrestrial climate change experiments and observational studies (ClimEx). Methods Ecol. Evol. 11, 22–37. https://doi.org/10.1111/2041-210X.13331

- ISO18400-102:2017, 2017. Soil quality -- Sampling -- Part 102: Selection and application of sampling techniques.
- Stenberg, B., Johansson, M., Pell, M., Sjödahl-Svensson, K., Stenström, J., Torstensson, L., 1998.
  Microbial biomass and activities in soil as affected by frozen and cold storage. Soil Biol.
  Biochem. 30, 393–402. https://doi.org/10.1016/S0038-0717(97)00125-9

---

## Author Response (AR1)

**Author's Response**

Our responses to each comment are in bold and below the original comment. Line numbers respond to the manuscript uploaded with track changes.

**Response to R1**

In this short communication the authors address the lack of evidence-based recommendations for storage of soil samples and extracts prior to analysis of organic and inorganic carbon (C) and nitrogen (N) and microbial biomass C and N. Based on literature and an online survey, the authors demonstrate that samples and extracts are stored in a multitude of ways. They further provide a case study in which they demonstrate the effects of sample and extract storage on measurements of organic, inorganic and microbial C and N, further stressing the need for standardisation of protocols.

The manuscript offers recommendations for sample and extract storage for the investigated methods, a flow-chart to guide researchers deciding on the most appropriate storage approach for their experiments and recommendations for reporting of the methods adopted for scientific publications. The results of the case study are a welcome reference for future research. In particular, the case study demonstrate that storage methods do not affect samples of different soil types (top vs subsoil) equally. This finding goes against the common assumption that storage affect all sample types similarly.

This short communication is relevant for the international science community, and fits well in the scope of SOIL. The introduction outlines the motivation of the objectives clearly, and the necessary details of the case study methodology and results are given in the supplementary material. The text is well written and pleasant to read. I have a few suggestions for corrections. Tables and figure are mostly supportive (particularly table 2). I have some suggestions to further clarify the text in tables 1 and 3. Also, I wonder whether the message the authors want to convey in section 4 with Table 4 isn't better served by integrating its information into the text. Lastly, I would ask the authors to consider including Fig S6 into the main text. I outline my general and minor comments in more detail below.

***Thank you very much for the positive comments and useful suggestions. We have detailed how we have addressed all the suggestions and recommendations proposed. See details below.***

General comments: The authors conducted an online survey and reflect on the outcome in L65 and further. Adding few lines about the nature of the survey and how representative the responses are would increase the value of its outcome in the authors argumentation.

***As recommended, we have included information on the survey data collection. The survey was conducted anonymously on Google surveys and promoted through Twitter a social media platform. A total of 68 participants provided information on how they typically store their soil and/or extract samples. Unfortunately, we do not have any survey participant demographics (e.g. types of institution, geography, positions of those who answered). Although this would have provided valuable insight as to who carries out specific storage methods and the representativeness of the responses, we decided that responses would remain anonymous as we felt that some researchers may not want to disclose how long samples may be left in storage prior to analysis.***

***We have added the survey questions below to supplementary materials:***

1) ***Realistically, how do you process your samples prior to carbon and nitrogen analysis (KCl, $K_2SO_4$, $H_2O$)?***
   i) ***Extract and run all my samples immediately***

ii)   *Extract immediately and store the extract until analysis*

iii)   *Store the soil and carry out extraction for immediate analysis*

iv)   *Store the soil and store the extract*

2)   *At what temperature do you generally store your extracts?*

i)   *4°C*

ii)   *-20°C*

iii)   *-80°C*

iv)   *Room temperature*

v)   *I do not store extracts prior to analysis*

vi)   *Other:*

3)   *At what temperature do you generally store your soil?*

i)   *4°C*

ii)   *-20°C*

iii)   *-80°C*

iv)   *Room temperature*

v)   *I do not store soil prior to analysis*

vi)   *Other:*

4)   *How long do you typically store your soils for prior to extraction?*

i)   *2 days or less*

ii)   *2 days - 1 week*

iii)   *1 - 3 weeks*

iv)   *3 weeks - 2 months*

v)   *Longer than 2 months*

vi)   *I do not store soil prior to analysis*

5)   *How long do you typically store your extracts for prior to analysis?*

i)   *2 days or less*

ii)   *2 days - 1 week*

iii)   *1 - 3 weeks*

iv)   *3 weeks - 2 months*

v)   *Longer than 2 months*

vi)   *I do not store extracts prior to analysis*

6)   *Any additional comments?*

Also, inclusion of Figure S6 into the introduction would aid the reader follow the narrative, or at least place it first in the Supplementary materials. For Figure S6, I recommend the authors to clarify the meaning of the colours, explain the abbreviation of RT, and consider scaling the size of the arrows to the % of cases.

*We have moved Fig S6 to the start of the supplementary material. We have also clarified the figure caption. The new figure caption reads: "Flow chart indicating the results of an online survey to determine how and for how long people store soils and/or extracts prior to analysis. Room temperature is abbreviated as RT. Survey date: December 2018 – June 2019. The proportion of people that followed each methodological step is indicated as 'n' in each arrow, which are also scaled in size depending on 'n'. The arrow colours correspond to the differences in soil sample storage, processing and extract storage prior to analysis. Pink: soil samples are extracted*

*immediately after collection, extracts are stored for later analysis. Blue: Neither soils nor extracts are stored. Green: soils are stored after collection, but extracts are analysed immediately after extraction. Yellow: both soils and extracts are stored."*

[Figure]

I miss the mentioning of other storage methods than those considered by the authors in their case study. L73 states the authors considered widely used storage methods, but it does not become clear to me how the selected methods compare to alternatives, for example drying. I recognize that a full evaluation is beyond the scope of this short communication yet adding a few examples could bring the proposed manuscript into a broader perspective. For example, storage and the achievability of soil samples are important considerations when choosing indicators for monitoring efforts (Ritz K, Black HIJ, Campbell CD, et al (2009) Ecol Indic 9:1212–1221. https://doi.org/10.1016/j.ecolind.2009.02.009).

*In response to the reviewer's comment, we have expanded on other storage alternatives in the introduction section to broaden the perspective of this short communication. We have added the following text to line 54 "In these cases, it is common practice to store samples for future analysis of which there are a broad range of storage practices. These include freeze drying, air drying, freezing and refrigerating samples and are typically chosen dependant on the analysis in question and time in which analysis can take place."*

*We have also added a short discussion as to why we chose to explore refrigeration and freezing treatments. From line 101, we have added the following text: "In our case study, we chose to explore refrigerating and freezing storage practices instead of other storage methods (e.g. air drying or freeze drying) because there is significant evidence to suggest that other methods are unsuitable for the variables we measure. For example, air drying soils has a strong effect on C and N pools, probably due to microbial death and nutrient release upon drying and rewetting (Jones and Willett, 2006; Kaiser et al., 2001; Li et al., 2012; Rolston and Liss, 1989). Additionally, freeze-drying is also known to have a strong effect on nutrient pools, as the chemical, physical, and physiological stresses inflicted by freeze-drying can kill soil microbes, releasing the microbial compounds into the soil (Islam et al., 1997)."*

.

*We would also like to highlight that our chosen storage treatments are commonly practiced as highlighted by the survey: 66% of participants store their soils at 4°C or -20°C whilst 87% of participants store their extracts at 4°C or -20°C.*

While sections 2 and 3 give helpful directions to soil scientists, I find section 4 less strong. Table 4 presents best reporting practices, yet the section does not mention why the listed requirements are important, nor gives a motivation for providing these recommendations. Also, I would recommend the authors to remove table 4, and instead list the four reporting recommendations in the text in section 4. To convey the best reporting practice, I don't think it is necessary to provide examples of poor reporting and giving examples of good reporting are easier to read when listed in the text.

*We agree have removed table 4 and integrated some of the information in the table into the text, line 330 - 338. This also highlights why it is so important to report storage methods in section 4.*

Minor issues:

L43: "that have considered these have taken into account few…" makes it easier to read

*This has been made clearer, line 65: "In many comparative studies exploring the impacts of methodological factors overlook soil and/or extract storage temperatures and duration, and those that have considered these, have taken into account few storage possibilities (Table 1)".*

Table 1: Column 6 "storage methods explores", remove the word 'only' in the listed examples.

*We have removed 'only' throughout the table.*

Table 1: Yolo loam: what soil type reference was used, e.g. WRB soil types?

*These are Typic Xerorthents, USDA classification.* https://soilseries.sc.egov.usda.gov/OSD_Docs/Y/YOLO.html

https://www.nrcs.usda.gov/Internet/FSE_DOCUMENTS/nrcs142p2_051232.pdf

Table 1: H2O – Rolson and Liss: at what temperatures were the soils stored frozen?

*We have amended this and added -10 °C*

Table 1: Plant available N – Soil Type: what is the difference between not applicable and not provided?

*We use the term "not applicable" for general recommendations that are made in the literature but are not based on comparative studies or investigations. We use the term "not provided" for comparative studies that do not describe the soils explored for different storage methods. We have made this clearer in the table caption, which has been revised to read line: "Summary of different recommendations for storage of soil or extract samples to measure soil nutrients found in the literature. This summary is non-exhaustive. The term "not applicable" under soil type refers to studies that were not based on comparative studies and therefore were not carried out on a soil type. The term "not provided" refers to comparative studies that do not describe the soils explored in the methods."*

Table 1: Plant available N – Jones & Willett 2006: can the text on soil type be shortened? F.e. "Unclear. All samples taken from temperate, oceanic locations";

*We have made this change.*

Table 1: Same row, last column: unclear what is meant with broad recommendation. What makes that a limitation?

*We have made this clearer. The recommendation made in the referenced publication states "store the extract for days". It is unclear how many days. We have changed this to "Vague recommendations made for extract storage length which could be open to different interpretation".*

L129: "suppress"

*We have change this to "halt" as suggested by reviewer S1.*

Table 3: Recommendation listed for Storage Methods is not formulated as a recommendation but reads like an observation.

*We agree with this point. As we have included more detail on alternative storage methods in the introduction, we have removed this row from Table 3.*

Table 3: Replicates-Heterogeneity: authors recommend 5 replicates, but it is unclear on which this is based. Statistically, number of required replicates depends on the variation within the group/treatment.

*We take this onboard and have addressed this issue by amending our recommendation in Table 3, this reads: "Generally, we recommend as many replicates as one can afford to have, but recommend no fewer than 4 as suggested by Jones and Willet (2006). For more guidance on choosing the number of replicates, we advise researchers to utilise the sample size calculator formula from Cochran and Cox (1957), p. 20. ".*

L155: "in the literature are"

*This has been corrected, line 329.*

Supplementary material: Table S2 is strangely outlined in the text and stands separately from its caption.

*We have ensured the position of the table caption is above the table*

Statistical analyses: how were normality and homogeneity assumptions checked?

*We have included this information to our statistical analysis in supplementary material. "Normality and homoscedasticity of the data were first checked using Anderson Darling and Levene's tests respectively."*

L99: unclear which variables were log transformed before analysis. Is this natural logarithm or Log10?

*We have included this information in the statistical analysis section of the supplementary material. "All variables were subjected to natural log transformation except for DOC which was not transformed."*

Fig S6: see in general comments

*We have moved Fig S6 to the start of the supplementary.*

**Response to R2**

The authors of this manuscript investigate the effects of storage method and duration on soil dissolved organic carbon (C) and nitrogen (N), and inorganic N. They identified common storage parameters

through a literature survey and an online survey, and then tested how these common storage methods affected recovery of extractable dissolved C and N. Based on this experiment they report recommendations for storage of soil samples. This manuscript is important to Soil science community and fits well within the scope of SOIL. The manuscript highlights the of standardization in storage parameters when measuring dissolved C and N. Furthermore, they found that different soils (subsoil vs topsoil) were affected differently by storage method. This manuscript was well written and pleasant to review. My major recommendations are related to the presentation of the case study data, and discussion of Soil storage methods.

***Thank you very much for your response. We have addressed all of your recommendations***

***We would like to clarify that the survey was actually carried out after our experiment. The purpose for the survey was to highlight the fact that researches do store soil samples and soil extracts, even though this information is usually not added into their methods. We also aimed to identify the proportion of researchers that carried out different soil storage methods.***

General comments:

I believe the manuscript as a whole would benefit from including the major findings from the case study into the main text. These results are currently included in the Results section of the supplement. Furthermore, when included in the main text, the major findings of the case study should be briefly summarized in the abstract. Section 1 clearly states the importance of the standardizing soil storage methods.

Section 2 The Case Study is the strongest evidence that there is a problem with non-standardized storage methods. Therefore, the results of the study should be included in the manuscript. The figures could remain supplementary since Table 2 summarizes the results. However, the Results should be moved to the main text with appropriate figure citations included in text.

***As suggested, we have expanded Section 2 by moving the results section from the supplementary into the main manuscript, please see line 141 to line 238. These lines include the description of the results obtained for DOC and DON measurements in water extracts , extractable ammonia and nitrate in KCl extracts , and microbial biomass carbon and nitrogen in K2SO4 extracts. We have kept the figures in the supplementary material and have referred to them in the main manuscript. The amended abstract that includes our case study findings now reads:***

***"It is widely accepted that the measurement of organic and inorganic forms of carbon (C) and nitrogen (N) in soils should be performed on fresh extracts taken from fresh soil samples. However, this is often not possible, and it is common practice to store samples (soils and/or extracts), despite a lack of guidance on best practice. We utilised a case study on a temperate grassland soil taken from different depths to demonstrate how differences in soil and/or soil extract storage temperature (4 °C or -20 °C) and duration can influence sample integrity for the quantification of soil dissolved organic C and N (DOC and DON), extractable inorganic nitrogen ($NH_4^+$ and $NO_3^-$), and microbial biomass C and N (MBC and MBN). The appropriateness of different storage treatments varied between topsoils and subsoils, highlighting the need to consider appropriate storage methods based on soil depth and soil properties. In general, we found that storing soils and extracts by freezing at -20 °C was least effective at maintaining measured values of fresh material, whilst refrigerating (4 °C) soils for less than a week for DOC/DON, up to a year for MBC/MBN, and refrigerating soil extracts for less than a week for $NH_4^+$ /$NO_3^-$ did not jeopardise sample integrity. We discuss and provide the appropriate tools to ensure researchers consider best storage practice methods when designing and organising ecological research involving assessments of soil properties***

*related to C and N cycling. We encourage researchers to use standardised methods where possible and to report their storage treatment (i.e. temperature, duration) when publishing findings on aspects of soil and ecosystem functioning. In the absence of published storage recommendations for a given soil type, we encourage researchers to conduct a pilot study and publish their findings."*

Line 93. Were the statistics performed in R? This should be stated and if the mixed effects models were performed using an R package that package should be cited. It appears they are cited in the supplement. That should be moved to the main text.

*As suggested, we have included this information in the main manuscript and reference the packages accordingly. Line 128 starts as: "All statistical analyses were carried out in R Version 3.6.1 (R Core Team, 2019)." And in line 132: "Mixed-effects models were performed for each measured variable with lme4 package (Bates et al., 2018) to test the effects of fixed factors …"*

Line 111-115 - Table 2: There appears to be a discontinuity between the table legend and the table itself. The table legend specifies that Red and Green squares, while the table has dark and light grey squares. It appears to be properly labelled in the note at the bottom of the table.

Line 112-113 - I am not following the statement from the legend, "Green denotes appropriate storage method with additional recommendations for storage length". Where are the additional recommendations? If they are in the main text I believe that should be specified. Line 113

*Thank you for highlighting this mistake. The legend and text have been corrected to read: "Table 2. Storage method recommendations for both temperate topsoil and subsoil. Dark grey denotes inappropriate storage methods for a specific analysis. Light grey denotes appropriate storage method, where storage length is annotated. Where storage length is annotated as 430 days we are unable to advise storage length beyond this due to the length of the experiment. Storage methods are deemed appropriate: 1) if the storage method does not compromise the sample integrity (defined as stored samples yielding soil parameter values within 20% similarity limits to fresh samples) for both topsoil and subsoils explored; and 2) where the same extractant type is used to measure different parameters, the storage method does not compromise the integrity of each parameter measured."*

Line 113 - I am also confused by this statement "Where we do not specify, stored samples did not differ from fresh samples through the entire experiment, 430 days." The only areas not specified are the dark grey boxes, which I understand to indicate an inappropriate storage method. Please clarify.

*We have removed this statement. We mistakenly didn't update the table caption accordingly when we changed the cells from being empty to then read <430 days.*

Additionally, if they did not differ during that time, then the sign should be greater than or equal to, or simply stated that recommendations beyond 430 days cannot be made due to the length of the experiment.

*We have corrected this. We have removed the < symbol from the table and state in the legend that we cannot advise storage beyond 430 day due to the length of our experiment (Please see amended caption above).*

Section 3 and 4 are strong. However, it appears to be implied that these recommendations are specific to studies comparing different soil, e.g. topsoil vs subsoil. Based on the study and the recommendations, it does not appear to apply to researchers carrying out an ecological study using a single soil that has been experimentally manipulated. If the authors agree with my conclusion then

they should state that their recommendations do not apply in the circumstance. If they do not, then I they should provide a strong argument for why not.

***We appreciate this remark and want to clarify that these recommendations would apply in the same way as if a study were using one soil type, because the accuracy of the measurements will still be affected by storage method. We have included an opening statement in section 3 to address this as follows, line 302: "The case study findings highlight how integral it is to consider best storage practice for soil analysis in any study/experiment, this includes studies exploring one or more soil types, site locations and/or treatment manipulations. We provide a step by step systematic flow chart to determine best storage methods for soil and soil extracts (Figure 1)." Please, see further details on this issue in the response to the short comments SC1 (below), issue 3, where we give a specific example and discuss why freezing is not a tolerable method for storage when studies are only looking at relative differences between treatments and/or site.***

Section 5 The authors should also address experimental manipulations of single soil types here as well.

***This point has been addressed in section 5. The table has been incorporated into text, line 331 to 338. We have clarified that one must still consider best storage practice when only working with one soil type.***

Technical notes

Line 19 space needed between "andNO3"

***This has been corrected, line 21.***

Line 31 I think this citation may be wrong in the citation manager because it does not have the author's name here or at the beginning of the citation in the references.

***We have checked and this reference is correct. ISO report standardised methods for laboratory practices and are usually referenced without an author name.***

Line 60 Add a space after the 4 and before the degree symbol

***This has been corrected.***

Table 1 Formatting to prevent last letter of some words moving to next line Line 180, I don't think the parentheses are necessary around 2020

***This has been corrected.***

**Response to SC1**

As a group of researchers from the Soil Science Department of the Research Institute of Organic Agriculture (FiBL), Switzerland, we discussed the manuscript entitled "Are researchers following best storage practices for measuring soil biochemical properties?" by Rhymes at al. 2020. Rhymes et al. 2020 raise the discussion on an important topic that concerns the whole soil science community. We would like to acknowledge the authors enormous work in a comprehensive and valuable case study on best practice storage conditions for soil samples and soil extracts for various commonly investigated biochemical parameters (with almost 2000 extractions performed). We highly appreciate their initiative in raising awareness on this vital, but often neglected topic and hope that their contribution will spark further work and exchange among soil scientists.

*Thank you for the positive response. We appreciate the evaluation your group have made and we respond to the comments individually.*

However, in our opinion some important aspects were not considered adequately and we have the following suggestions for improvement:

1) The data on which Rhymes et al. base their guidance should be provided in the main manuscript rather than the supplementary information (SI). While the authors themselves claim that "[. . .] optimal storage conditions will vary across different soils and ecosystems" (Line 75), but also between top- and subsoil, as shown by their own case study, they come to very generalized recommendations on best practice storage in Table 2, which we find contradictory. To us it is not quite clear how the authors come to their recommendations, or at least some differentiation is lacking. For instance, in Figures S4a and S5a, storing frozen extracts up to 430 days seems tolerable for both MBC (only topsoil) and MBN (both top- and subsoil), but in Table 2 freezing extracts for assessment of microbial biomass is indicated as completely inappropriate.

*As recommended, we have expanded Section 2 by moving the entire results section from the supplementary into the main manuscript, please see line 141 to line 238. The figures have remained in the supplementary and have been referred to throughout the main manuscript.*

*Summarising our findings into generalised recommendations was a particularly difficult task considering the differences we found between soil depths. We also want to highlight that these are not guidelines that should be followed if using different soil types or soils at different depths to those explored. The general rational for the guidelines summarised in Table 2 were that: 1) a storage method must have deemed appropriate for both subsoils and topsoils and 2) where the same extractant is used to measure different parameters they too must deem appropriate for both sets of parameters. In the example you give for MBC and MBN, we would like to highlight that storing frozen extracts was not deemed appropriate for measuring MBC in subsoils and therefore did not meet our first clause. Furthermore, microbial biomass is generally measured as microbial biomass nitrogen and carbon and would use the same extraction solution to measure both and consequently does not meet our second clause.*

*To clarify this issue, we have added this statement to line 228 to 230: "However, freezing samples did not significantly affect the concentration of N in fumigated or unfumigated samples, and thus frozen extracts was a suitable storage method to measure MBN."*

*We have also clarified our rational for making such guidelines in the table caption, line 256. This reads: "Table 2. Storage method recommendations for both temperate topsoil and subsoil. Dark grey denotes inappropriate storage methods for a specific analysis. Light grey denotes appropriate storage method, where storage length is annotated. Where storage length is annotated as 430 days we are unable to advise storage length beyond this due to the length of the experiment. Storage methods are deemed appropriate: 1) if the storage method does not compromise the sample integrity (defined as stored samples yielding soil parameter values within 20% similarity limits to fresh samples) for both topsoil and subsoils explored; and 2) where the same extractant type is used to measure different parameters, the storage method does not compromise the integrity of each parameter measured."*

*We do also provide some guidance on this issue in Table 3 bottom row "deciding on the best storage practice" but feel that with the added information in the caption this is much clearer.*

2) The discussion of changes upon storage should be further elaborated and put in context with existing literature (e.g. the literature reviewed in Table 1). For example, Stenberg et al. (1998) suggest

that soils can be stored frozen for up to 13 months for assessing microbial biomass, while Rhymes et al. recommend not to freeze soil at all for any kind of biochemical analysis they considered in their manuscript. How would the authors explain these differences?

*We have expanded the discussion and have compared our results with those published in the literature (see below for more details). For example, the key difference between our study and Stenberg et al. is that they only measured MBC, whilst we measured both MBC and MBN. In line with the findings of Stenberg et al. we also found that freezing soils to measure MBC was acceptable (up to 75 days for topsoil and 430 for subsoil), but not for MBN (although acceptable for topsoil up to 430 days). As discussed above, due to subsoil sample integrity being jeopardised by freezing soils to quantify MBN we deemed this storage method inappropriate. Please see line 195, this now reads "We found that freezing soils to measure MBC was acceptable (up to 75 days for topsoil and 430 for subsoil), but not for MBN (although acceptable for topsoil up to 430 days). We therefore deemed freezing soils as an inappropriate storage method for quantifying microbial biomass because the subsoils were jeopardised by freezing soil samples to quantify MBN. Our recommendations are therefore contrary to those made by Stenberg et al. (1998), despite finding similar results for MBC."*

*Another potential explanation for these differences could be differences in soil microbial communities. They worked with soils from Upsala, Sweden that experience marginally lower average winter temperatures than the soils we collected. In turn, the microbial communities in their soils could be more adapted to colder temperatures causing them to respond differently to freezing storage methods.*

In Line 130, Rhymes et al. speculate about microbial processes as the main driver of changes in stored soils or extracts and they suggest storing samples under conditions which suppress microbial activity completely. Given the major changes still happening in frozen extracts over time for NO3 (Figure S3a), do the authors suggest that freezing is not suppressing microbial activity sufficiently? Could there be other mechanisms responsible for this trend?

*We do not explore the mechanisms responsible for the changes to samples under different storage conditions and therefore can only discuss potential mechanisms. Indeed, freezing does suppress activity, but our results demonstrate that this is not enough to maintain sample integrity. As a result, (on line 235) we recommend other potential storage options that completely halt microbial activity such as the acidification of extractants. To avoid confusion, we have changed "completely suppress microbial activity" to "completely halt microbial activity".*

*In response to freezing not supressing microbial activity, there is evidence for this; freezing has been shown to decrease microbial biomass as a result of damaged cell structures (Černohlávková et al., 2009). As a result microbes that are not damaged by the freezing process profit from the organic molecules made available from the proportion of microbes that have died (Stenberg et al., 1998). This could be a potential mechanism to explain the shift over time whilst in the freezer but we are unable to conclude this as we did not investigate the mechanisms.*

The discussion currently provided in the SI should be moved to the main manuscript in order to increase its visibility to the scientific community.

*We agree with the reviewer's comments and have included some of the potential mechanisms for the sample deterioration we observed under different storage conditions. We have moved this drom the supplementary, this can now be found in the new Section 2.3 – Discussion.*

3) The importance of the underlying research questions is neglected: The authors only look into relative changes in the measured parameters in comparison to freshly extracted and immediately analyzed samples. However, many studies aim at investigating relative differences between treatments rather than obtaining absolute data on fresh samples. In fact, appropriate storage conditions are not only part of the method, but also depend strongly on the research question. In many cases, standardized pretreatments (for example pre-incubation of soil after refrigerated storage for microbial N and C), freezing of all samples before extraction etc. might produce smaller errors than immediate extractions, where differences upon sample collection, transport, outside temperature upon sampling etc. would arguably cause bigger effects than the storage treatment. With this regard, especially the change in the measured parameters upon prolonged duration of storage is relevant. For instance, Rhymes et al. consider freezing of soil or extract for analysis of NH4 inappropriate (Table 2 or Figure S3b), however, changes here seem to appear immediately upon freezing, with marginal changes thereafter (Figure S3b). For studies only interested in relative differences between treatments or sites, freezing thus would be a tolerable storage method. Again, we think that the recommendations should be more differentiated and take potential research questions into account.

*We agree that other procedures for sample collection and preparation will also affect sample integrity. We refered to these other procedures in the introduction to ensure readers also consider other aspects that will impact sample integrity. These sentences have been added to the introduction from Line 47 to address this issue: "It is therefore integral that researchers consider each factor that can impact accurate and reliable analytical measurements, which can include sampling procedures (e.g. strip removal of turf), transport (e.g. transport length and temperature), storage (e.g. temperature), preparation for analysis (e.g. sieving mesh size and when samples are sieved) and analytical methods (e.g. temperature, shaking times and filter types). Here we focus solely on sample storage……. In these cases, it is common practice to store samples for future analysis. These can include freeze drying, air drying, freezing and refrigerating samples, and the method is typically chosen dependent on the analysis in question and time in which analysis can take place."*

*However, we disagree with the reviewers' view that freezing is a tolerable method for storage when studies are only looking at relative differences between treatments and/or site. Our findings demonstrate that storage can impact soil from different depths differently, which implies that soils of different types are likely to respond differently to storage methods. We did not explore the mechanisms for the differences observed and therefore cannot assume that this is correlated to soil depth alone. It might be as a result of other factors such as differences in nutrient status or microbial populations for example. If this were to be true, two samples of the same soil type with different nutrient statuses would respond differently to a soil storage method. Or two soils under different experimental treatments that had affected their nutrient status or microbial communities would also respond differently. We therefore do not recommend that researchers assume that samples from the same site will respond the same to a storage method and that the relative differences between treatments will not change. Additionally, not all variables respond in the same way to storage, and if researchers are to study C/N ratios for example, storage effect will have a strong impact on them. Finally, it is true that there are some changes that occur immediately and some that occur over storage time. However, this immediate change was not equal in the two soil depths studied (for example frozen extracts for NH$_4$ determination, or freezing soil for DON), and hence, the same logic applies.*

*We have now included this in our discussion to address this issue, line 269: "It is commonly assumed that any changes to soil biochemistry from storage methods will occur equally for all samples. Here,*

*we provide evidence to show that changes do not occur equally which could have major implications for the findings of ecological studies. We did not investigate the mechanism behind different responses of two soils to the storage treatments, but it could be related to many factors such as differences in nutrient status or microbial communities. As a result, any treatment that affects soil properties have the potential to also affect the response of soils to storage. Even if sample biochemistry changes immediately as a result of storage but subsequently remains stable over storage time, in our study this effect varied between the two soil depths. Therefore, even if the research question is to compare between treatments applied to the same soil type, strict storage limits should be followed. We suggest that all samples should be stored under the same conditions that allow the preservation of samples from the soil type, site and/or treatment with the highest sensitivity to storage."*

4) From our own experience, but also highlighted by the results of the survey which Rhymes et al. conducted amongst different laboratories (note that the documentation on how the survey was conducted could be expanded), storage of both soil and soil extracts are common practice. This is owed to the mere impossibility to collect, extract and analyze samples in one day, especially with high sample numbers or when sample collection has to be conducted at large spatial distance to the lab. In this context, we find their conclusion on "appropriate" or "inappropriate" storage too general. How about defining an acceptable relative error, e.g. by handling the samples in one way or the other? Furthermore, as indicated above, relative errors occurring immediately (e.g. upon freezing or un-freezing) should be distinguished from continued changes upon prolonged storage.

*We agree that we needed to provide more information about our survey approach. The survey was conducted on Google surveys and promoted through twitter a social media platform. We have now included the survey questions in the supplementary materials (see details in response to referee 1). Line 98 now reads "*survey (details provided in Supplementary 3.4)"*. The rational for the survey and the survey questions are now included in the supplementary material.*

*Regarding the "appropriate" or "not appropriate" recommendations, they are already taking into account and accepting a 20% error for each storage treatment explored. But if researchers consider that a higher % error is acceptable for a particular study due to strong logistical limitations, a higher error to determine appropriateness could be used. Regarding changes happening immediately upon storage, please, refer to our response above.*

5) With their study, Rhymes et al. made an important point on the effect of storage conditions, but we miss the broader picture. The discussion should expand also on other aspects potentially compromising the integrity of soil samples, such as sampling procedure, transport, pre-treatments or handling of the samples in the laboratory. We believe that the whole soil science community should put more effort into defining common standards and evaluating potential errors during the whole procedure from sample collection, transportation and storage until analysis. Comparing the effect of storage conditions with the effects of these other aspects would help to identify sources of major errors and design experiments accordingly.

*We agree that the manuscript is strengthened with a comment to address other aspects of sample collection and processing beyond storage methods can also impact sample integrity. We have addressed this in your 3rd comment (please see 3 above) and have also included an amended sentence in section 5 Conclusions, line 360*

*"We stress that researchers must also consider other practices beyond just storage (e.g. sieving samples, transport, extraction procedures) as each methodological step between sample collection*

*and analysis can introduce errors to measurements that are intended to be field representative. We encourage researchers to utilise standardised methods where possible (see e.g. Halbritter et al. (2020)) and to follow best storage practices for specific soil types to allow reliable comparison of data from different studies."*

6) If each group has to carry out their own pilot studies and resulting storage conditions will vary substantially, then meta-analyses will become even more difficult than they are now. Besides, the recommendations for such pilot studies would need to be really concise, e.g. how many time points would need to be analyzed? It would be important to learn as much as possible from the experiment conducted by Rhymes et al. As an alternative to pilot studies, why not put an effort into identifying suitable reference materials that can be included in each study?

*We believe that if more people that publish and report their pilot studies, this will generate the data required to conduct a large meta-analysis that paints a much clearer picture as to how storage impacts our results. This could potentially result in a standardised storage protocol being produced. But to date we know very little about the impacts of storage methods. In the supplementary material "Extended material and methods", we provide all the details necessary to replicate our study. Furthermore, in Table 3, we consider the different aspects that researchers should take into account when designing their own pilot studies. We do not suggest or think that it should be necessary that all the pilot studies should be an exact replica of our study. It is more appropriate that they focus on the aspects (e.g. storage methods, length, similarity limit) that can better inform their own experiments. Indeed, storage conditions will vary amongst these pilot studies but high variability in ecological studies already exist too and are commonly used for meta-analysis. Efforts should still be made despite this hurdle. We believe this commentary will spark the necessary discussions amongst the soil community to give sample storage more consideration than it currently does.*

*Suitable reference materials may be an option for quality control purposes. However, this would need to be explored for future recommendation as little is known about the mechanisms behind the shifts associated with storage and most importantly whether reference materials would behave similarly to living soils.*

In addition to these general thoughts, here are some more detailed comments:

Sampling procedure and soil sample preparation

- While we understand the reasoning behind their sampling approach (topsoil sampled three weeks after sampling the subsoil), in most of our experiments this is simply not an option, e.g. due to distant sampling locations and the importance of a uniform sampling time point.

*This approach was due to logistical constraints, and we do not suggest that researchers stagger their sampling times for different soils routinely.*

- Soil samples were taken in June. Would results be different if soils had been sampled in winter or at a different initial water content? Generally speaking, the effect of seasonality should be discussed.

*We agree that seasonality will affect any soil biochemical results measured. It is unknown whether they would also respond differently to storage methods.*

*We appreciate that we do not make note of this in the manuscript. We have now added this sentence in the discussion to address this comment, line 281: "We would like to note that due to the high temporal variability that the temperate soils explored experience, there is the potential that storage*

*methods could impact sample integrity differently depending on when the samples were collected. Understanding the mechanisms responsible for jeopardising sample integrity under different storage methods will help determine the best storage methods for the time in which samples are collected (e.g. season), soil type and depth."*

- Apparently, Rhymes et al. use "field replicates" for their extractions (SI Line 43ff): Soil was sampled from five locations (transect over the field with 10m distance between plots) in 0.5 x 0.5 m pits. These replicates were later on used for the extraction/different storage treatments. This sampling approach explains the high data variability upon the individual time points and storage conditions and should have been discussed by the authors.

*We agree that true field replicates always result in higher variability, but we used this approach to avoid pseudo replication, and to provide a more robust representation of variability within our field experiment. We calculated relative change to help account for high variability. Additionally, we included a 20% similarity limit, instead of a stricter 10% similarity limit, which also helps account for this high variability.*

*We have included a statement on this issue in the supplementary methods: "This approach of sampling and keeping separate true field replicates was chosen to avoid pseudo replication, and to properly represent the high variability associated with typical ecological field experiments. High data variability was accounted for by calculating relative change for each individual replicate compared to corresponding fresh samples, and by increasing the acceptable similarity limit to 20% (see statistical analyses for details)."*

*We consider that adding that information in the methods is sufficient and we do not feel it is necessary to include a statement in the discussion.*

- The time of sieving/homogenization was not investigated, since Rhymes et al. sieved all soil samples on the day after sample collection and stored all the soils sieved. Would the results have been different if soils had been sieved only after storage, immediately before extraction? Extraction procedure and handling of extracts

*Yes, this will certainly yield different results, we address this comment both in the introduction and conclusion (Please see our response to points 3 and 5). We do not feel that it is necessary to discuss in too much detail as the purpose of this study was to look at storage methods. We feel that with the inclusion of the sentences we propose in the introduction and the conclusion we have made it much clearer that researchers need to consider all of these aspects including sieving, from sieving mesh size to when the soils are sieved immediately after collection or just before analysis.*

- SI Line 71: For K2SO4 extraction, no blanks were performed. While this seems valid for the calculation of microbial C and microbial N as difference between fumigated and non-fumigated extracts, we find this problematic for reporting values on total C and total N in both fumigated and non-fumigated extracts, which were not corrected for blanks (compare Figures S4 b, c and Figure S5 b, c)

**We agree that our results would have benefit from $K_2SO_4$ blanks. When the experiment was designed, we did not consider the possibility of presenting the raw unfumigated and fumigated data, and only the microbial C and N, deeming unnecessary to process and store $K_2SO_4$ blanks (which would have increase the number of extracts to an additional 180 samples). However, for the purpose of this study, we are only looking at relative differences between storage treatments and feel that presenting the raw unfumigated and fumigated data without blank correction is acceptable. In accordance, the recommendations we make are based on the microbial C and N**

**values and not the unfumigated and fumigated values. The fumigated and unfumigated samples only helped in the discussion to explain some of the effects observed.**

- The molarities of extractants (K2SO4, KCl) are not reported throughout the whole manuscript.

*This has been added throughout. The molarity used was 0.5M for $K_2SO_4$ and 1M for KCl.*

- Scaling of extractions procedure: Authors report that 5 g of moist soil were extracted. This is a very low amount considering any potential inhomogeneity in the soil. Due to the high number of replicates (n=5) this might be acceptable.

*This is very common practice and is reflected in numerous publications, some even use less soil (e.g. 2.5g of soil to 25ml extractant in Jones and Willett).*

Additionally, soil moisture content (e.g. between top- and subsoil) was ignored upon extraction, which might lead to differences in the soil-to-solution ratio. Equal amounts of dry soil equivalents should be used for a standardized extraction procedure.

*Although we did not utilise equal amounts of dry soil equivalents, we did correct for differences in soil weights as a result of differences in soil moisture for final nutrient concentration calculations. It has come to our attention that this is not appropriately detailed in our methods and we have amended this in the supplementary material. We would like to highlight that it is common practice to correct for the difference in dry weight when calculating concentrations, please see the standardised protocol in Halbritter et al. (2020). We would also like to draw your attention to Table S1 and note that we only saw a 2% variation in soil moisture and therefore feel that this will not have impacted our results.*

- Scaling could also be added as another point to consider for a pilot study within Table 3 (extraction methods; recommendation: do not up-/down-scale the used amounts but use the same amounts as planned for the main experiment).

*We already elude to this concept in table 3, under replicates, pseudoreplication "Do not store soils or extracts in bulk. The same weight or volume of soil or extract must be stored separately for each storage treatment and time point." But we agree that it should include the idea of scaling down. We modified this and have created another row in the table for scaling, line 320:*

*"Consideration: Scaling Issues: pseudoreplication, reproducibility. Recommendations: Do not scale your soils or extracts for storage up (bulk storage) or down. The same weight or volume of soil or extract must be stored separately for each storage treatment and time point as the one planned for the main experiment."*

- Freezing and un-freezing procedures were not investigated as further factors. From our experience, it makes a difference in which position extracts are frozen (e.g. vertical or horizontal placement of tubes) and under which conditions extracts or soils samples are thawed (e.g. thawing soils over night at 4◦C or extracting frozen soil immediately with the solution).

*This is an interesting point and would make for a good exploratory study. We have added details on the specific conditions in which extracts were frozen in the supplementary extended methods (they were all frozen vertically). Details on how soils and extracts were thawed are already in the supplementary information, extended methods.*

*We have added the idea of the potential effects of freeing/thaw procedures on measurements in the discussion, line 229: "Due to the potential for freeze-thaw cycles to impact sample biogeochemistry (Černohlávková et al., 2009) it is important to consider and be consistent with the*

*freeze/thaw procedure, such as the position in which extracts are frozen (vertical or horizontal placement of tubes) or under which conditions extracts or soils samples are thawed (e.g. thawing soils over night at 4°C or extracting frozen soil immediately with the solution)."*

Statistics/Figures/Data presentation

SI Line 104: Why did the authors use a plot digitizer to extract numeric data from their own plots?

*We define sample integrity as unusable when the confidence interval of the predicted model intercepts the outline of the similarity limit which is shaded in grey. We didn't consider it was important as to how we calculated this and chose what seemed to be the easiest way. This can of course be calculated through the predicted confidence intervals from our models, which has yielded the same results. We have amended this accordingly in the supplementary methods.*

- Figure S1b: In the figure caption, authors indicate that there was a technical problem with the DON measurement on the last time point (Day 430) and thus, data should not have been included. However, in the figure there a data points also for this sampling time.

*Graph S1b is correct. However, you have brought to our attention that the 430 day sampling point was not visible on all other graphs due to the x axis only going up to 6 rather than 6.06 (log of 430 days). This is a graphical plotting error rather than a statistical error, whereby no sampling points were excluded from out the linear mixed models. We have amended all the graphs in the supplementary material. As an example, here is our corrected graph forS4a, which now includes the last time point.*

[Figure]

- For some of the analyzed parameters, the replicates show a very high data variability. However, this seems not always represented in the confidence interval displayed (e.g. Figure S3 a: NO3 values for frozen extracts vary widely, while the confidence interval seems to be very small).

*We would like to highlight that these are 95% upper and lower confidence intervals for the predictive fitted ratio change values based on the mixed effects models carried out. This is not an individual confidence interval at each time point, and that is the reason in this instance why data variability is high, but the represented confidence interval is small.*

- In Table 3, authors recommend to use twice the number of replicates for the baseline (freshly extracted and analyzed samples). However, for their own case study they did not follow this recommendation or at least did not report it.

*We did not use twice the number of replicates, but upon reflection for such a study we think this is valuable and have therefore added it into the recommendations.*

Technical comments:

- Typo in Table 1: Plant available N, reference "Jones and Willett 2006", under storage methods explored it should probably be -18◦C

*This has been corrected.*

- Line 60: Wording is misleading. Stenberg et al. 1998 also sieved the soil prior to storing it at different temperatures.

*We have removed the word sieved so it is no longer misleading, line 92.*

- Table 2: There seems to be a mistake in the table header. We do not see any red or green squares. We assume that the information given below the table ("Dark grey denotes inappropriate storage method and light grey appropriate.") gives the same information?

*We have corrected this and have added grey colour information to the table caption. This reads, line 255:*

*"Table 2. Storage method recommendations for both temperate topsoil and subsoil. Dark grey denotes inappropriate storage methods for specific analysis. Light grey denotes appropriate storage method, where appropriate storage length is annotated. Where storage length is annotated as 430 days we are unable to advise storage length beyond this due to the length of the experiment."*

- Line 135: Figure 1 should only have a figure caption below, but not additionally above.

*This has been corrected, line 305.*

- Table S2: Table header is missing.

*This has been corrected. Table S2 header: "Table S2 Summary of extraction methods used in this experiment"*